# TDP1 suppresses chromosomal translocations and cell death induced by abortive TOP1 activity during gene transcription

Diana Rubio-Contreras [1,2] & Fernando Gómez-Herreros [1,2] ✉

DNA topoisomerase I (TOP1) removes torsional stress by transiently cutting one DNA strand. Such cuts are rejoined by TOP1 but can occasionally become abortive generating permanent protein-linked single strand breaks (SSBs). The repair of these breaks is initiated by tyrosyl-DNA phosphodiesterase 1 (TDP1), a conserved enzyme that unlinks the TOP1 peptide from the DNA break. Additionally, some of these SSBs can result in double strand breaks (DSBs) either during replication or by a poorly understood transcription-associated process. In this study, we identify these DSBs as a source of genome rearrangements, which are suppressed by TDP1. Intriguingly, we also provide a mechanistic explanation for the formation of chromosomal translocations unveiling an error-prone pathway that relies on the MRN complex and canonical non-homologous end-joining. Collectively, these data highlight the threat posed by TOP1-induced DSBs during transcription and demonstrate the importance of TDP1-dependent end-joining in protecting both gene transcription and genome stability.

DNA topoisomerases are essential enzymes with critical functions in DNA metabolism[1]. These enzymes release the torsional stress generated in the DNA by a wide variety of processes such as transcription and replication, facilitating DNA transactions. Type IB topoisomerases (i.e., human DNA topoisomerase I) relax superhelical stress by generating DNA single strand breaks (SSBs) that allow rotation of one broken DNA strand relative to the other intact strand. DNA topoisomerase 1 (TOP1) is thought to be particularly relevant during the maintenance of genome stability and during transcription. In fact, TOP1 participates in several steps of transcription by RNA polymerase II (RNAPII), from initiation to termination, highlighting its relevance for the regulation of gene expression[2].

A key intermediate of TOP1 activity is the cleavage complex, in which the DNA is cleaved and the enzyme is covalently bound to the 3' end of the DNA through of a phosphotyrosine linkage[3]. TOP1 cleavage complexes (TOP1ccs) are normally transient since the topoisomerase reseals the break at the culmination of its catalytic cycle. However, DNA metabolism related processes or the presence of antitumor agents that act as topoisomerase poisons can stabilize TOP1ccs, prolonging the half-life of this intermediary. These situations can lead to the formation of irreversible TOP1ccs, commonly known as 'abortive', that represent a threat to genome integrity. To be repaired, abortive TOP1ccs are first ubiquitinated, marking them for degradation by the proteasome[4]. Not only the proteasome but other proteases can participate in this debulking step[4]. Proteolyzed TOP1 peptide remains covalently bound to the 3' end of the break and reveals a 5'-hydroxyl moiety that triggers break signalling by PARP1 and the recruitment of XRCC1 and associated SSB repair factors. Tyrosyl DNA phosphodiesterase 1 (TDP1), a highly conserved enzyme among eukaryotes, surgically removes the TOP1 adduct by hydrolyzing the phosphotyrosine bond that links it to the DNA. Canonical 3'-hydroxyl/ 5'-phosphate ends are restored by polynucleotide kinase

[1]Instituto de Biomedicina de Sevilla, IBiS, Hospital Universitario Virgen del Rocío/CSIC/Universidad de Sevilla, 41013 Seville, Spain. [2]Departamento de Genética, Universidad de Sevilla, 41012 Seville, Spain. ✉e-mail: fgomezhs@us.es

3′-phosphatase (PNKP), and repair is completed by ligation of DNA ends by DNA ligase III (LIG3)[5]. Defects in several SSB repair factors result in hereditary neurological syndromes, among them, SCAN1, characterized by cerebellar ataxia, neuropathy and seizures, is caused by a homozygous mutation in TDP1[6,7].

During S-phase, collision of the replisome can convert an abortive TOP1cc into a single-ended DSB. DSBs are the most cytotoxic lesion of those occurring in DNA, hence the important role of the selective TOP1 poison camptothecin (CPT) and its derivatives in chemotherapy[8]. Intriguingly, TOP1-induced SSBs can be converted into DSBs in the absence of replication[9]. These replication-independent DSBs, which are associated with TOP1 activity in transcription, can be induced with CPT[10,11]. These breaks have been classically rationalized as abortive TOP1ccs in proximity to other pre-existing lesions on the opposite strand. However, more recently, it has been shown that several nucleases can also promote these breaks since concomitant TOP1cc trapping and R-loop resolution by XPF, XPG or FEN1, or the activity of MUS81 on an abortive TOP1ccs are additional sources of TOP1-induced DSBs[12,13]. Altogether, these and other studies demonstrate that replication-independent TOP1-induced DSBs arise from multiple sources and are very likely heterogeneous.

Notably, while the exact origin of replication-independent TOP1-induced DSBs is not fully understood, how these breaks are repaired remains almost unknown. This is a pivotal question since transcription-associated DSBs are a significant and poorly understood endogenous source of genome instability in eukaryotic cells. In fact, DSBs that arise during gene transcription can result in chromosome translocations, key events at the origin and development of many solid tumours and leukemias[14]. Moreover, TOP1-induced DSBs might be unrecognised contributors to neurodegenerative syndromes associated with defects in SSB repair[10,11]. Most studies to date support that homologous recombination repair (HR) represents the main TOP1-induced DSB repair pathway, both of replication-dependent and independent origin. However, HR is not present in G1 or quiescent cells, suggesting that, at least in these cases, a second DSB repair pathway should be repairing such lesions. In agreement, TDP1-defective quiescent lung carcinoma cells have been shown to be impaired in TOP1-induced DSB repair[13] and TDP1 activity seems dispensable for initiating HR. In addition, in the absence of TDP1, DNAPKcs, a key regulator of the canonical non-homologous end joining repair pathway (cNHEJ), is hyperactivated[12].

Here we have directly addressed the repair of replication-independent TOP1-induced DSBs while studying their impact in transcription-associated genome instability. We show that TOP1 removal by TDP1 is a key step of the TOP1-induced DSB repair pathway. This repair pathway is not totally dependent on ligase IV (LIG4) and is independent of DNA polymerase theta (POLQ), core members of cNHEJ and theta-mediated end joining (TMEJ), respectively. In addition, we show that TDP1-dependent DSB repair suppresses genome translocations and cell death resulting from these breaks, which we revealed as a source of genome rearrangements associated with transcription in mammalian cells. Finally, we demonstrate that TOP1ccs can be processed by MRE11 and, in TDP1 deficient cells, DSB repair is completed by cNHEJ, resulting in increased genome instability and cell death. The clarification of this TDP1-dependent repair pathway has important implications in understanding SSB-associated neurodegenerative disease, transcription-associated genome instability and the efficacy and mutagenic effects of TOP1 poison-based chemotherapy.

## Results

### Transcription-associated TOP1-induced DSBs are partially repaired by cNHEJ

To study replication-independent TOP1-associated DSBs we synchronised diploid human RPE-1 hTERT cells in G0/G1 by confluency and serum starvation (for details, see Materials and Methods)[15]. Over 98% of cells were arrested in G0/G1 and replicating cells were rarely detected in culture (Fig. S1). Exposure to CPT rapidly induced SSBs in quiescent cells, measured by the alkaline comet assay (Fig. 1a). CPT also induced 53BP1 and H2AX serine 139 phosphorylation (hereafter γH2AX) immunofoci, common markers of DSBs[16] (Fig. 1b and Fig. S2a). Concomitantly, TOP1 poisoning repressed global transcription (Fig. S2b), and further SSBs and DSBs accumulation rapidly ceased (Fig. S2c, d). Of note, the kinetics of DSBs formation lagged behind that of the SSBs, suggesting that TOP1-induced DSBs may derive from TOP1-induced SSBs. RNAPII elongation inhibition with 5,6-dichloro-1-β-D-ribofuranosylbenzimidazole (DRB) before TOP1 poisoning resulted in up to 80% reduction in both SSBs and DSBs (Fig. 1a, b). The dependency of TOP1-induced DSBs on RNAPII transcription was confirmed with the catalytic inhibitor α-amanitin (Fig. 1b). In contrast, inhibition of RNA polymerase I (RNAPI) with CX5461 caused no significant reduction in 53BP1 foci in response to CPT treatment (Fig. 1b), indicating that RNAPI transcription had no substantial influence on the formation of TOP1-induced DSBs. These results demonstrate that TOP1 activity associated with RNAPII transcription can be a prominent source of replication-independent SSBs and DSBs in quiescent RPE-1 cells.

We then measured replication-independent TOP1-induced DSB repair rates by following the kinetics of 53BP1 foci after CPT removal in quiescent cells. Repair of TOP1-induced DSBs was completed in 3 h after CPT removal (Fig. 1c). The loss of LIG4 significantly delayed TOP1-induced DSB repair, suggesting that cNHEJ is involved in the repair of these DSBs (Fig. 1c). However, the repair defect observed in *LIG4*[−/−] cells was not total. Repair during the first hour was similar to the control and after 3 h 60% of breaks were already repaired (Fig. 1c). This result was unexpected since we have previously shown that the loss of LIG4 completely abolishes the repair of other DSBs associated with transcription, such as those induced by the abortive activity of DNA topoisomerase II (TOP2)[15]. Indeed, *LIG4*[−/−] cells were completely deficient in the repair of DSBs induced with the TOP2 poison etoposide; after 6 h of repair more than 80% of DSBs remained (Fig. 1d). These results suggest that, in addition to cNHEJ, other alternative repair pathways may deal with TOP1-induced DSBs.

To study the possible implication of pathways alternative to LIG4, we inhibited either RAD52, a key factor in the single strand annealing repair pathway (SSA) that is also active in non-cycling cells[17], and RAD51, a key factor in HR. Despite HR is restricted to S and G2 phases, RAD51 has been found associated to TOP1-induced DSBs in G1[18]. Neither RAD52 nor RAD51 inhibition affected the repair rate of TOP1-induced DSBs in quiescent cells, suggesting that neither SSA nor HR are involved in the repair of these breaks (Fig. S3a, b). Finally, we impaired TMEJ by knocking out POLQ, an essential factor of this pathway[19], by using CRISPR-Cas9 tools (Fig. S3c). *POLQ*[−/−] cells exhibited hypersensitivity to the TOP2 poison etoposide, in agreement with a previous study[20] (Fig. S3d). However, POLQ deficiency did not affect TOP1-induced DSB repair, suggesting that TMEJ is not involved in the repair of these breaks in quiescent cells (Fig. S3e). Since TMEJ has been proposed to operate as a backup to cNHEJ[21], to formally discard its implication we treated control and *LIG4*[−/−] cells with novobiocin (NVB) or ART558, two potent POLQ inhibitors that block TMEJ[22,23]. Notably, POLQ inhibition did not promote any further repair defect, suggesting that TMEJ is not involved in the repair of TOP1-induced DSBs in the absence of proficient cNHEJ (Fig. 1e, f). Taken together, these results suggest that replication-independent TOP1-induced DSBs can be repaired by cNHEJ and by undefined DSB repair mechanisms.

### TOP1-induced DSBs are repaired by a TDP1-dependent repair pathway

Next, we wondered whether TDP1, with a known function in TOP1-induced SSB repair might be also involved in the repair of TOP1-

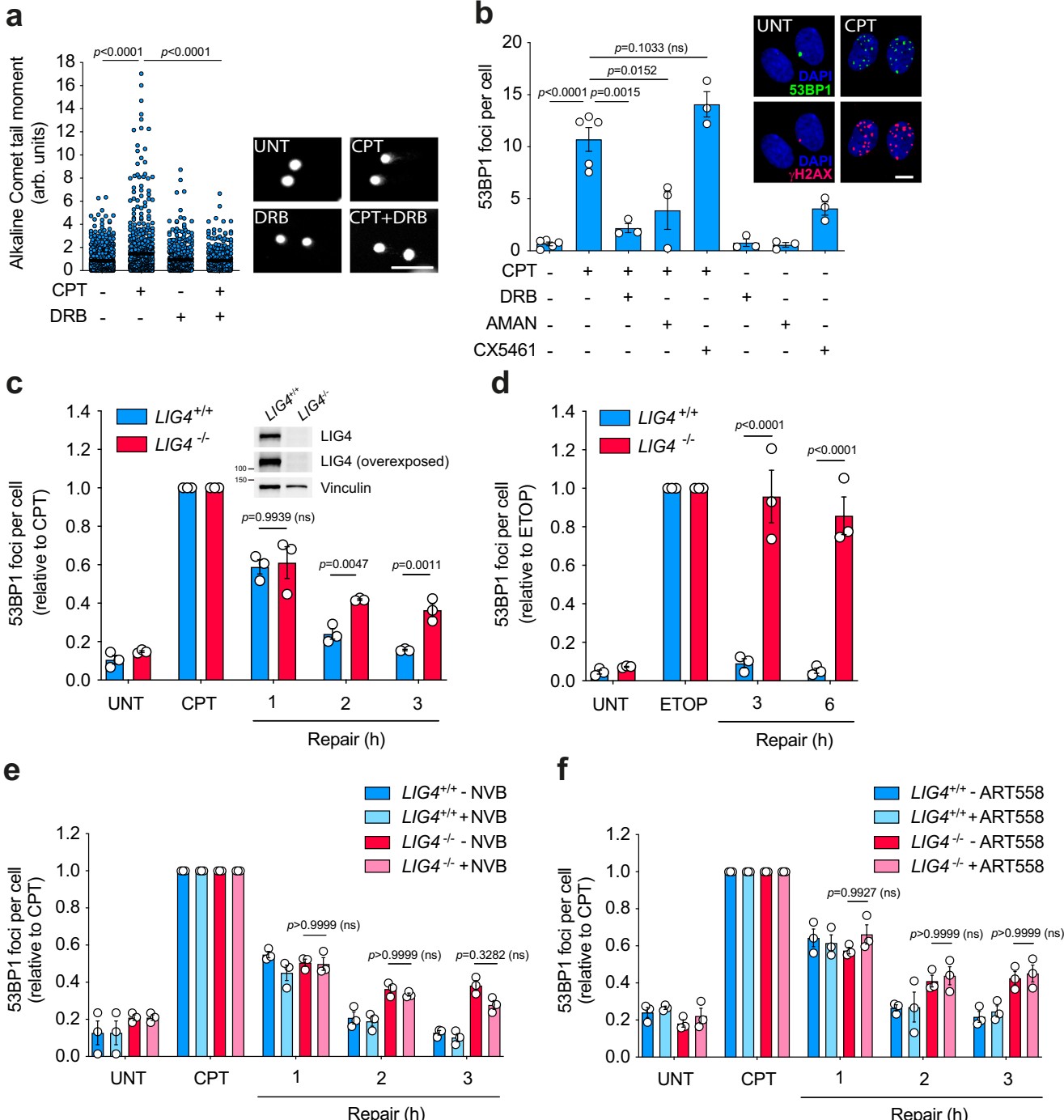

**Fig. 1 | Transcription-associated TOP1-induced DSBs are repaired by cNHEJ.**
**a** Detection of DNA breaks by the alkaline comet assay in serum-starved RPE-1 cells treated with CPT (25 μM) for 1 h. Where indicated, cells were pre-incubated with DRB (100 μM) for 3 h prior to CPT treatment. From left to right: $n = 736$, $n = 734$, $n = 527$ and $n = 427$ cells over 3 independent experiments. Representative images of nuclei are shown. **b** 53BP1 foci in serum-starved RPE-1 cells treated with CPT (25 μM) for 1 h. Where indicated, cells were pre-incubated with DRB (100 μM), α-amanitin (AMAN) (10 μM) or CX5461 (20 μM) for 3 h prior to CPT treatment. $n \geq 3$ independent experiments. Representative images of 53BP1 foci (green), γH2AX foci (red) and DAPI counterstain (blue) are shown. **c**, **d** 53BP1 foci in serum-starved $LIG4^{+/+}$ and

$LIG4^{-/-}$ RPE-1 cells after 1 h treatment with 12.5 μM CPT (**c**) or 20 μM etoposide (ETOP) (**d**), and during repair in drug-free medium. $n = 3$ independent experiments. Protein blot of LIG4 is shown in **c**. **e**, **f** 53BP1 foci in serum-starved $LIG4^{+/+}$ and $LIG4^{-/-}$ cells after 1 h treatment with 12.5 μM CPT, and during repair in drug-free medium. Where indicated, cells were incubated with the POLQ inhibitors novobiocin (NVB) (100 μM) (**e**) or ART558 (10 μM) (**f**) during repair. $n = 3$ independent experiments. UNT = untreated. Data were represented as mean ± SEM. Statistical significance was determined by two-tailed unpaired $t$-test for **a** and **b**, and by two-way ANOVA followed by Sidak's multiple comparisons test for **c**–**f**. ns non-significance. Scale bar, 100 μm for **a** and 10 μm for **b**. Source data are provided as a Source Data file.

induced DSBs. We employed RPE-1 cells in which we disrupted *TDP1* using CRISPR-Cas9 (Fig. 2a). *TDP1*$^{-/-}$ cells exhibited hypersensitivity to CPT (Fig. 2a). Notably, upon CPT treatment, these cells accumulated very high levels of SSBs measured by the alkaline comet assay

(Fig. 2b) and nuclear poly ADP-ribose (hereafter PAR) (Fig. 2c). The specificity of PAR signal was checked by PARP1 inhibition (Fig. 2c). Remarkably, *TDP1*$^{-/-}$ cells accumulated very high levels of DSBs induced by CPT, suggesting that replication-independent TOP1-

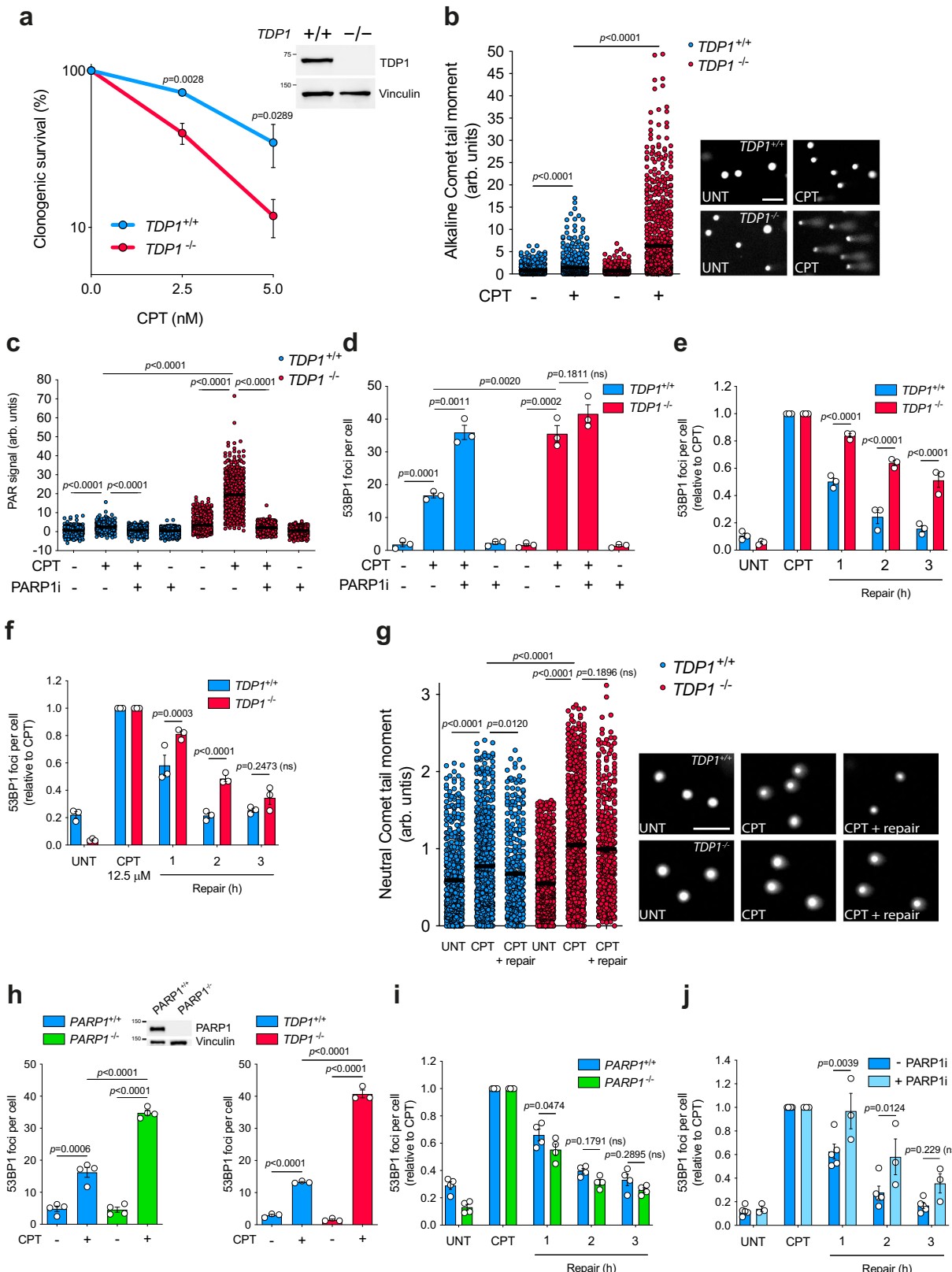

induced DSBs might arise from the accumulation of unprocessed TOP1-induced SSBs and/or that *TDP1*−/− cells are unable to efficiently repair them (Fig. 2d). In fact, TOP1-induced DSB accumulation was also promoted when CPT treatment was combined with PARP1 inhibition (Fig. 2d). Strikingly, *TDP1*−/− cells exhibited a strong delay in the repair of TOP1-induced DSBs, suggesting that TDP1 is involved in the

repair of these breaks (Fig. 2e). This repair defect was also observed at a lower CPT concentration (Fig. 2f), ruling out that it was an artefact due to the underestimation of the number of DSBs at the time of induction in *TDP1*−/− cells. Additionally, TOP1-induced DSB formation and repair were directly measured by the neutral comet assay, that specifically evaluates DSBs[24], confirming that *TDP1*−/− RPE-1 cells

**Fig. 2 | Transcription-associated TOP1-induced DSBs are repaired by a TDP1-dependent repair pathway. a** Clonogenic survival of *TDP1*[+/+] and *TDP1*[−/−] RPE-1 cells treated with CPT for 48 h. *n* = 3 independent experiments. Protein blot of TDP1 is shown. **b** Detection of DNA breaks by alkaline comet assay in serum-starved *TDP1*[+/+] and *TDP1*[−/−] cells treated with CPT (25 μM) for 1 h. From left to right: *n* = 736, *n* = 734, *n* = 417 and *n* = 771 cells over 3 independent experiments. **c** Quantification of PAR (immunofluorescence) in serum-starved *TDP1*[+/+] and *TDP1*[−/−] cells treated with CPT (25 μM) for 2 h. Where indicated, cells were pre-incubated with PARP inhibitor KU58948 (1 μM) for 1 h prior to CPT. From left to right: *n* = 648, *n* = 756, *n* = 667, *n* = 687, *n* = 511, *n* = 652, *n* = 572 and *n* = 670 cells over 2 independent experiments. **d** 53BP1 foci in serum-starved *TDP1*[+/+] and *TDP1*[−/−] cells treated with CPT (25 μM) for 1 h. Other details as in **c**. *n* = 3 independent experiments. **e, f** 53BP1 foci in serum-starved *TDP1*[+/+] and *TDP1*[−/−] cells after 1 h treatment with CPT (25 μM) (**e**) or (12.5 μM) (**f**), and during repair in drug-free medium. *n* = 3 independent

experiments. **g** Detection of DSBs by neutral comet assay in serum-starved *TDP1*[+/+] and *TDP1*[−/−] RPE-1 cells treated with CPT (25 μM) for 1 h and after 2 h repair in drug-free medium. From left to right: *n* = 481, *n* = 621, *n* = 331, *n* = 764, *n* = 840 and *n* = 370 cells over 3 independent experiments. **h** 53BP1 foci in serum-starved wild-type, *PARP1*[−/−] or *TDP1*[−/−] cells treated with CPT (12.5 μM) for 1 h. *n* = 3 independent experiments. **i** 53BP1 foci in serum-starved *PARP1*[+/+] and *PARP1*[−/−] cells after 1 h treatment with CPT (12.5 μM), and during repair in drug-free medium. *n* = 4 independent experiments. **j** 53BP1 foci in serum-starved RPE-1 cells incubated with the PARP inhibitor KU58948 (1 μM) during repair. *n* = 3 independent experiments. UNT untreated. Data were represented as mean ± SEM. Statistical significance was determined by two-tailed unpaired *t*-test for **b**–**d**, **g** and **h** and by two-way ANOVA followed by Sidak's multiple comparisons test for **a**, **e**, **f**, **i**, **j**. ns non-significance. Representative images of nuclei are shown in **b** and **g**. Scale bar, 100 μm for **b** and **g**. Source data are provided as a Source Data file.

accumulate CPT-induced DSBs and are unable to efficiently repair them (Fig. 2g).

Since TDP1 recruitment to TOP1cc is partially dependent on PARP1 activity and these proteins are epistatic for the repair of abortive TOP1ccs[25], we analysed CPT-induced DSB formation and repair in *PARP1*[−/−] cells[26]. Notably, PARP1 deficiency promoted the accumulation of high levels of DSBs similarly to *TDP1*[−/−] cells (Fig. 2h). However, it did not provoke a significant delay in the repair of TOP1-induced DSBs, demonstrating that the repair defect observed in the absence of TDP1 was not an artefact of the high accumulation of DSBs upon CPT treatment (Fig. 2i). Contrary, inhibition of PARP1 after CPT removal, and thus not inducing accumulation of DSBs due to unprocessed SSBs, resulted in a significant repair defect, specially at early times (Fig. 2j). Next, we measured CPT-induced levels of abortive TOP1ccs by TOP1-conjugated DNA isolation by in vivo complex of enzyme (ICE) assay[27] and subsequent immunoblotting using a TOP1cc specific antibody[28]. As previously described, TDP1 deficiency promoted a very high accumulation of TOP1ccs (Fig. S4)[10]. This accumulation was also observed after PARP1 inhibition and in *PARP1*[−/−] cells, although to a lesser degree (Fig. S4)[29]. Overall, these results suggest the requirement of abortive TOP1cc removal, and demonstrate a TDP1 dependency, in the process of TOP1-induced DSB repair.

## TDP1-dependent DSB repair is backed up by cNHEJ

To characterize this TDP1-dependent DSB repair pathway, we next explored the genetic relationships with the three known DNA repair ligases. We achieved more than 90% depletion of LIG3 with CRISPR-Cas9 by combining two single guide RNAs (sgRNA) (Fig. 3a). LIG3 depletion generated a significant increase of DSBs upon CPT treatment, in agreement with our previous results in *TDP1*[−/−] cells (Fig. 3a). However, in contrast to the defect observed in *TDP1*[−/−] cells, LIG3 depletion did not provoke a significant defect in the repair of these breaks, suggesting that TOP1-induced DSBs are repaired by a ligase other than LIG3 downstream of TDP1 (Fig. 3b) and confirming that SSB repair defects promotes the formation of DSBs. These results were further confirmed with two independent *LIG3*[−/−] clones (Fig. S5a). Strikingly, depletion of LIG3 in *TDP1*[−/−] cells did not provoke any further defect in TOP1-induced DSB repair (Fig. 3b).

Next, we studied DNA ligase I (LIG1), the participation of which in a SSB repair subpathway and in TMEJ has been previously described[5,30]. LIG1 depletion did not increase CPT-induced DSBs, suggesting that the implication of LIG1 in TOP1-induced SSB repair is, if any, minimal (Fig. 3a). More importantly, LIG1 depletion did not provoke a significant delay in TOP1-induced DSB repair either (Fig. 3c). These results were further confirmed with two independent *LIG1*[−/−] clones (Fig. S5b). Depletion of LIG1 in *TDP1*[−/−] cells did not provoke any additional defect in TOP1-induced DSB repair (Fig. 3c).

Redundancy of LIG3 and LIG1 has been observed both in SSB repair and TMEJ. To elucidate whether LIG1 and LIG3 participate in alternative routes in TOP1-induced DSB repair, we depleted LIG1 by

siRNA in control and *LIG3*[−/−] cells (Fig. 3d). Notably, LIG1 depletion did not strongly affect TOP1-induced DSB repair (Fig. 3d). These results suggest that if LIG1 and LIG3 are implicated in TOP1-induced DSB repair, they must be redundant with a third ligase. This led us to test whether LIG4 could participate in TOP1-induced DSB repair mediated by TDP1. For this, we depleted TDP1 in *LIG4*[−/−] cells. TDP1 depletion increased DSB accumulation in wild-type and *LIG4*[−/−] cells (Fig. 3e). Strikingly, TDP1-depleted *LIG4*[−/−] cells exhibited a synergistic repair defect of TOP1-induced DSBs, that extended further than 4 h, suggesting that TDP1 and LIG4 participate in different TOP1-induced DSB repair pathways (Fig. 3f). Consistent with this result, inhibition of DNAPKcs impaired TOP1-induced DSB repair in *TDP1*[−/−] cells compared to control cells (Fig. 3g). Altogether these results demonstrate that, in the absence of TDP1, TOP1-induced DSB repair relies on cNHEJ. Notably, the repair defect detected in TDP1-depleted *LIG4*[−/−] cells was not observed in *TDP1*[−/−] *POLQ*[−/−] cells (Fig. S5c). Similarly, RAD52 inhibition did not affect TOP1-induced DSB repair rates in *TDP1*[−/−] cells (Fig. S5d).

To confirm the participation of cNHEJ in TOP1-induced DSB repair in cells lacking TDP1, we analysed the co-localization of XRCC4, the main partner of LIG4, and the DSB marker γH2AX by proximity ligation assays (PLA) in XRCC4-V5 expressing quiescent RPE-1 cells (Fig. 3h). An increase in XRCC4-V5-γH2AX PLA foci was observed in wild-type cells subjected to increasing concentrations of CPT. Of note, this effect was markedly higher in *TDP1*[−/−] cells (Fig. 3h), indicating that cNHEJ involvement in TOP1-induced DSB repair is enhanced in the absence of TDP1. Altogether, these results demonstrate that those TOP1-induced DSBs that are not repaired by TDP1, are repaired by cNHEJ in quiescent RPE-1 cells.

## TDP1 suppresses genome instability and cell death induced by TOP1

To estimate the physiological relevance of TDP1-dependent DSB repair, we addressed the impact of TOP1-induced DSBs on genome stability by analysing chromosomal translocations. Wild-type and *TDP1*[−/−] cells were maintained in G0/G1 during, and 6 h after treatment with CPT, until repair was completed (Fig. 4a). In fact, only after 2 h of repair, PAR signal was back to untreated conditions in TDP1-lacking cells (Fig. S2e). Then, cells were released into cell cycle to enable the detection of translocations in metaphase (Fig. 4a and Fig. S6a). Despite wild-type cells accumulating up to 20 DSBs after 2 h of high dose CPT treatment (Fig. S2d), chromosomal translocations were barely detected, indicating that translocations resulting from this type of DSB are rare (Fig. 4a). Strikingly, *TDP1*[−/−] cells treated with CPT exhibited a dramatic increase of this type of chromosomal rearrangements compared to wild-type cells (Fig. 4a). Moreover, translocations associated with TDP1 loss were prevented by pre-incubation with DRB before drug treatment, confirming that RNAPII-mediated gene transcription was the source of TOP1-associated chromosomal translocations (Fig. 4b). PARP1 deficiency and PARP1 inhibition during CPT treatment also increased translocations (Fig. 4c), in agreement with the higher

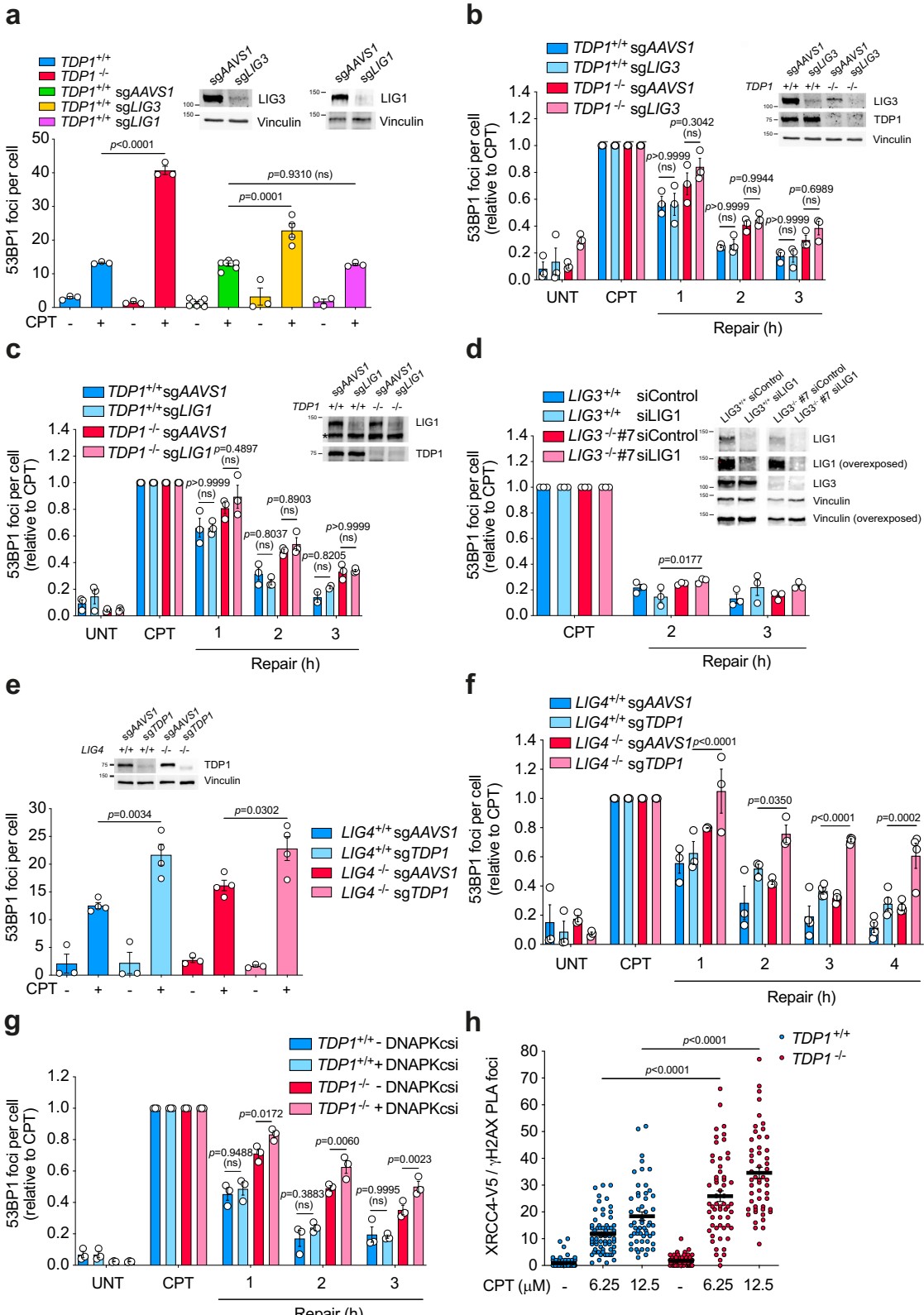

number of DSBs that *PARP1*⁻/⁻ cells and PARP inhibitor induced upon CPT treatment (Fig. 2d, h). It is worth noting that, although the level of DSBs generated by CPT in *TDP1*⁻/⁻, *PARP1*⁻/⁻ and wild-type RPE-1 cells treated with PARP inhibitor was similar (Fig. 2d, h), translocations were significantly much more frequent in *TDP1*⁻/⁻ cells, excluding the possibility that the increase in translocations was a mere consequence of an increase in DSBs.

To formally confirm that CPT induced reorganisations were replication-independent and not formed by replisome encounters with unrepaired SSBs, we directly examined the frequency of chromosome translocations in quiescent cells using premature chromosome condensation (see materials and methods and Fig. S6b, c for details). After CPT treatment and repair, quiescent wild-type and *TDP1*⁻/⁻ cells were fused to HeLa mitotic cells promoting the

**Fig. 3 | TDP1-dependent DSB repair is backed up by cNHEJ. a** 53BP1 foci in serum-starved *TDP1*[+/+] mock-depleted (sg*AAVS1*), LIG3-depleted (sg*LIG3*) or LIG1-depleted (sg*LIG1*) and *TDP1*[−/−] RPE-1 cells treated with CPT (12.5 µM) for 1 h. *n* = 3 independent experiments. Protein blots of LIG3 and LIG1 are shown. **b, c** 53BP1 foci in serum-starved *TDP1*[+/+] and *TDP1*[−/−] mock-depleted (sg*AAVS1*) or LIG3-depleted (sg*LIG3*) (**b**) or LIG1-depleted (sg*LIG1*) (**c**) cells after 1 h treatment with 12.5 µM CPT, and during repair in drug-free medium. *n* = 3 (**b**) and *n* ≥ 2 (**c**) independent experiments. Protein blots of TDP1, LIG3 and LIG1 are shown. Vinculin was blotted in an independent membrane (**b**). Asterisk indicates a non-specific band. **d** 53BP1 foci in serum-starved mock-depleted (siControl) or LIG1-depleted (siLIG1) *LIG3*[+/+] and *LIG3*[−/−] RPE-1 cells. *n* = 3 independent experiments. **e** 53BP1 foci in serum-starved mock-depleted (sg*AAVS1*) or TDP1-depleted (sg*TDP1*) *LIG4*[+/+] and *LIG4*[−/−] RPE-1 cells after 1 h treatment with 12.5 µM CPT. *n* ≥ 3 independent experiments. **f** 53BP1 foci in serum-starved mock-depleted (sg*AAVS1*) or TDP1-depleted (sg*TDP1*) *LIG4*[+/+] and *LIG4*[−/−] cells after 1 h treatment with 12.5 µM CPT, and during repair in drug-free medium. *n* ≥ 3 independent experiments. **g** 53BP1 foci in serum-starved *TDP1*[+/+] and *TDP1*[−/−] cells. Where indicated, cells were incubated with the DNAPKcs inhibitor NU7441 (10 µM) during repair. *n* = 3 independent experiments. **h** PLA assay showing XRCC4-γH2AX proximity in serum-starved *TDP1*[+/+] (XRCC4-V5) and *TDP1*[−/−] (XRCC4-V5) cells treated with CPT for 2 h. PLA assays were performed using rabbit α-V5-tag antibody and mouse α-γH2AX antibody. From left to right: *n* = 76, *n* = 70, *n* = 54, *n* = 58, *n* = 58 and *n* = 52 cells over 2 independent experiments. UNT untreated. Data were represented as mean ± SEM. Statistical significance was determined by two-tailed unpaired *t*-test for **a**, **e** and **h** and by two-way ANOVA followed by Sidak's multiple comparisons test for **b**–**d**, **f** and **g**. ns non-significance. Source data are provided as a Source Data file.

condensation, and thus the visualization, of single chromatid chromosomes in quiescent cells (Fig. S6b, c). TOP1-induced DSB formation was directly measured by visualization of small chromosomal fragments upon CPT treatment and by Giemsa staining, confirming that *TDP1*[−/−] cells accumulate CPT-induced DSBs (Fig. S6b). Strikingly, chromosomal translocations severely increased in *TDP1*[−/−] cells compared to wild-type cells after repair (Fig. 4d and Fig. S6c). These results demonstrate that TOP1-induced DSBs can be a prominent source of chromosomal translocations and that TDP1 prevents them.

To clarify the role of the TDP1-dependent DSB repair pathway in protecting genome stability, we next studied chromosomal rearrangements induced by CPT in LIG1- and LIG3-depleted cells. LIG1 and LIG3 depletion promoted a very modest increase of translocations compared to *TDP1*[−/−] cells but higher than wild-type cells (Fig. 4e). Comparing the increase of DSBs observed in *TDP1*[−/−] and in LIG3-depleted cells (Fig. 3a), these results further confirm that the cause of the formation of CPT-induced chromosomal rearrangements is TDP1 deficiency and not simply a higher DSB accumulation. Contrary to the other ligases, *LIG4*[−/−] cells did not show any increase in translocations compared to control cells (Fig. S6d). Notably, in agreement with the absence of a TOP1-induced DSB repair defect observed under POLQ deficiency, POLQ inhibition in *TDP1*[−/−] cells and *TDP1*[−/−]*POLQ*[−/−] cells did not significantly change the accumulation of chromosomal translocations (Fig. S6e). These results indicate that TDP1 suppresses the formation of chromosomal translocations induced by TOP1 during transcription while alternative pathways promote them.

Finally, we wanted to evaluate how CPT-induced DSBs affected cell viability in quiescent cells. CPT-induced cell death in asynchronous cells strongly depends on active DNA replication. However, to study the contribution of transcription-associated DSBs to CPT toxicity, we studied clonogenic survival of cells that had been treated with CPT, allowed to repair while quiescent and finally transferred to serum-containing medium (Fig. 4f, see details in materials and methods). Strikingly, *TDP1*[−/−] cells exhibited a high sensitivity to CPT (Fig. 4f). Of note, CPT hypersensitivity was prevented by pre-incubation with DRB, suggesting that TDP1 is essential to survive to TOP1-induced DSBs arising during transcription. Importantly, this suppression was not restricted to *TDP1*[−/−] cells, but also observed in wild-type RPE-1 cells (Fig. 4f). These results demonstrate that TDP1 prevents toxicity promoted by TOP1-induced DSBs associated with transcription in quiescent cells.

## MRE11 corrupts TOP1-induced DSBs repair

Our results reveal that *TDP1*[−/−] cells are not fully defective in TOP1-induced DSB repair, pointing to the existence of TDP1-independent pathways. DNA end processing of blocked DSBs is a dark box in which there is a high degree of redundancy. We decided to focus on MRE11 nuclease within the MRE11-RAD50-NBS1 (MRN) complex, since it has been shown to participate in trimming a wide variety of DNA ends in G1[31]. Additionally, MRE11 can cleave the 3′-phosphotyrosyl bond of an abortive TOP1cc within DNA[32]. To study the possible role of MRE11

in replication-independent TOP1-induced DSB repair, we first analysed the localization of endogenous MRE11 to abortive TOP1ccs by PLA in quiescent cells. Notably, MRE11 and abortive TOP1cc proximity was promoted by CPT in *TDP1*[−/−] cells (Fig. 5a). Next, to evaluate whether MRE11 can remove abortive TOP1ccs in vivo, we measured CPT-induced levels of abortive TOP1ccs. Notably, inhibition of MRE11 endonuclease activity by PFM01, but not inhibition of MRE11 exonuclease activity by PFM39, resulted in a subtle but significant accumulation of abortive TOP1ccs (Fig. 5b). As expected, *TDP1*[−/−] cells accumulated higher levels of abortive TOP1ccs, but this accumulation was still significantly higher when MRE11 endonuclease activity was inhibited (Fig. 5b). These results suggest that endonucleolytic activity of MRE11 is partially responsible for the processing of abortive TOP1ccs in RPE-1 quiescent cells, and that it operates in a pathway alternative to TDP1.

Next, to analyse the participation of MRE11 in TOP1-induced DSB repair we repeated repair kinetics using nuclease inhibitors. Strikingly, inhibition of endonucleolytic activity of MRE11 promoted a significant delay in the repair of TOP1-induced DSBs in wild-type and *TDP1*[−/−] cells, suggesting that MRE11 is involved in the repair of these breaks in a TDP1 alternative pathway (Fig. 5c). Notably, no effect was observed with PFM39 (Fig. 5c).

Given the role of TDP1 in preventing genome instability and cell death induced by CPT, we next evaluated the effect of inhibiting the nuclease activity of MRE11 on chromosomal translocations induced by CPT in TDP1-deficient cells. Inhibition of MRE11 endonuclease activity but not the exonuclease activity resulted in a partial but significant reduction in chromosomal translocations (Fig. 5d). Importantly, this result was confirmed by direct depletion of MRE11 expression by using specific siRNAs (Fig. S7a). Notably, neither PFM01, PFM39, nor MRE11 depletion negatively affected transcription, discarding an indirect effect on TOP1 activity and SSB formation (Fig. S7b, c). Altogether, these results suggest that TOP1cc processing by the endonucleolytic activity of MRE11 can lead to chromosomal translocations.

To determine the physiological relevance of MRE11-associated TOP1-induced DSB repair we next studied the genotoxicity induced by CPT in quiescent cells. Notably, preventing MRE11 endonucleolytic activity significantly reduced genotoxicity of CPT in *TDP1*[−/−] cells (Fig. 5e). Taken together, these results indicate that MRE11, through its endonuclease activity, can act on TOP1-mediated intermediates, giving rise to DSBs that can lead to genomic instability.

## Discussion

Abortive cycles of DNA topoisomerases are a constant threat to genome integrity. In this study, we have addressed the repair of replication-independent TOP1-induced DSBs. RNAPII transcription inhibition before CPT treatment almost ablated DSB formation, corroborating that these replication-independent DSBs are associated with TOP1 activity during transcription[10,11]. These results agree with the observed reduction of 53BP1 foci in G1 upon TOP1 poisoning in HeLa cells pre-incubated with DRB[33].

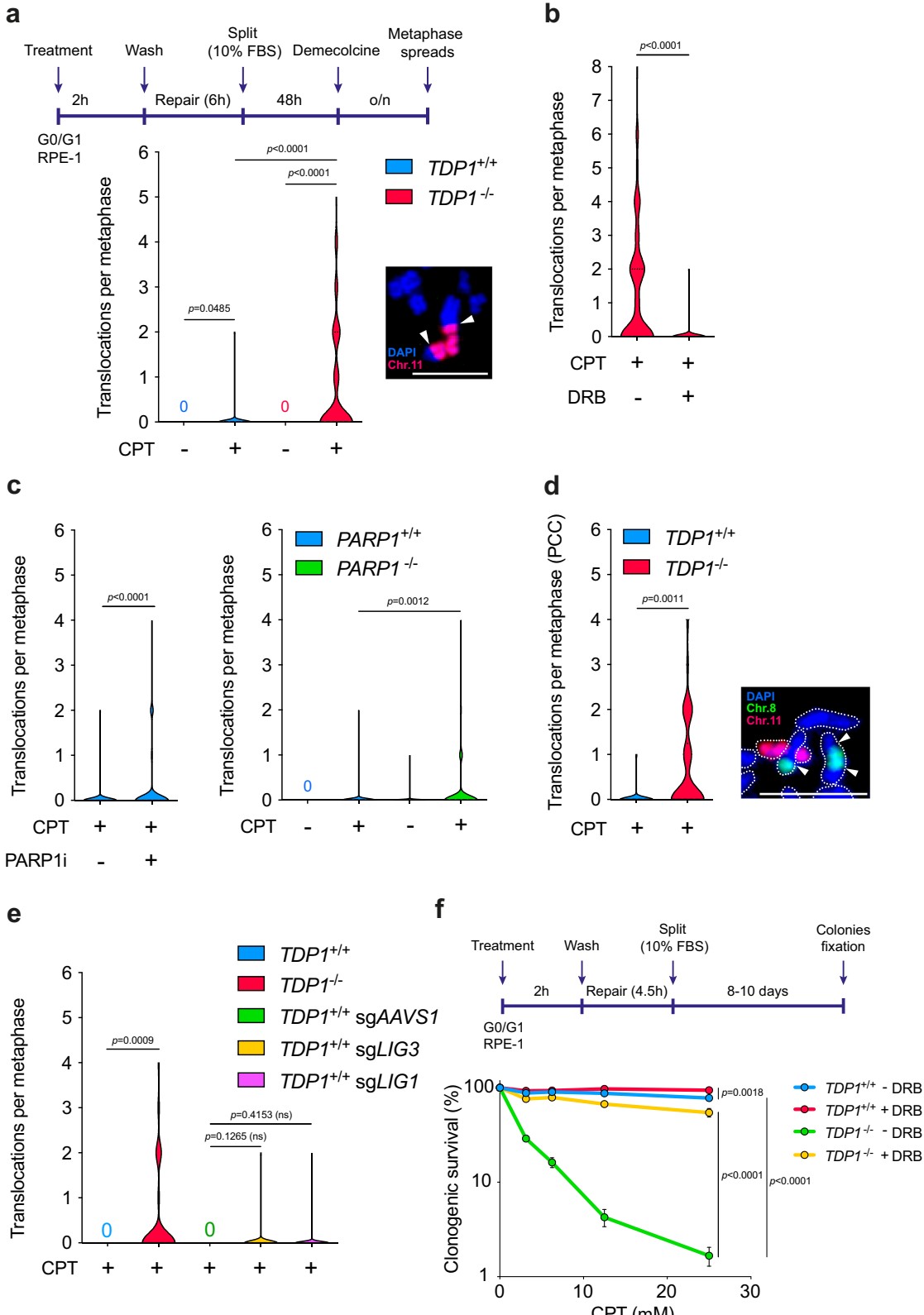

Our DSB repair kinetic analyses revealed the implication of LIG4 in the repair of CPT-induced DSBs. However, despite the fact that cNHEJ is the main DSB repair pathway in mammalian cells, we did not observe a repair defect in CPT-treated $LIG4^{-/-}$ quiescent cells as strong as the one detected after treatment with the TOP2 poison etoposide. The different DSB repair phenotypes highlighted the particularity of TOP1-induced DSBs and led us to explore alternative repair routes. Notably,

we failed to detect any repair defect inhibiting RAD52, POLQ or RAD51, suggesting that these core factors of SSA, TMEJ and HR, respectively, may not be necessary for the repair of TOP1-induced DSBs in quiescent cells. Our repair kinetics showed that $TDP1^{-/-}$ cells exhibited a strong defect in CPT-induced DSB repair, in agreement with a previous report[13]. These results suggest that TDP1, and thus the removal of the TOP1cc-DNA adduct, is required to initiate TOP1-induced DSB repair

**Fig. 4 | TDP1 suppresses TOP1-induced chromosomal translocations and cell death. a**–**c** Translocation frequencies were quantified in serum-starved wild-type, *TDP1*[−/−] or *PARP1*[−/−] RPE-1 cells in metaphase spreads prepared 48 h after CPT treatment (25 μM) for 2 h followed by 6 h repair in drug-free medium. Where indicated, *TDP1*[−/−] cells were pre-treated with DRB (100 μM) for 3 h (**b**) and *PARP1*[+/+] cells with PARP inhibitor KU58948 (1 μM) for 1 h (**c**) prior to, during, and 6 h after CPT treatment. From left to right: $n = 200$, $n = 508$, $n = 200$ and $n = 193$ cells over 3 independent experiments for **a**, $n = 93$ and $n = 98$ cells over 2 independent experiments for **b**, $n = 357$ and $n = 408$ cells over 3 independent experiments for **c**, left and $n = 200$, $n = 200$, $n = 201$ and $n = 190$ cells over 2 independent experiments for **c**, right. Workflow and a representative image of chromosomal translocations are shown in **a**. White arrows indicate translocation events. **d** Translocation frequencies were quantified in serum-starved *TDP1*[+/+] and *TDP1*[−/−] cells treated with CPT (25 μM) for 2 h followed by 24 h repair in drug-free medium and fused with HeLa cells synchronized in metaphase. From left to right: $n = 30$ and $n = 34$ cells

over 2 independent experiments. A representative image of chromosomal translocation is shown. White arrows indicate translocation events. Dotted lines delimit chromosomal outlines. **e** Translocation frequencies in serum-starved *TDP1*[+/+] mock-depleted (sg*AAVS1*), LIG3-depleted (sg*LIG3*) or LIG1-depleted (sg*LIG1*) and *TDP1*[−/−] cells. From left to right: $n = 100$, $n = 88$, $n = 100$, $n = 250$ and $n = 150$ cells over 2 independent experiments. Other details as in **a**. **f** *Top*, workflow. *Bottom*, clonogenic survival of serum-starved *TDP1*[+/+] and *TDP1*[−/−] cells treated with CPT for 2 h, and after 4.5 h repair in drug-free media. Where indicated, cells were pre-treated with DRB (100 μM) for 3 h prior to CPT treatment. After repair, cells were collected and re-cultured in serum containing media. $n = 3$ independent experiments. Data were represented as mean ± SEM. Statistical significance was determined by two-tailed unpaired *t*-test for **a**–**e** and by two-way ANOVA followed by Sidak's multiple comparisons test for **f**. Scale bar, 10 μm for **a** and **d**. ns non-significance. Source data are provided as a Source Data file.

(Fig. 5f). Interestingly, depletion of TDP1 in *LIG4*[−/−] cells revealed a synergistic defect in CPT-induced DSB repair suggesting that TDP1 and LIG4 can participate in independent repair pathways. We also considered the possibility that LIG1 and LIG3 complete TDP1-dependent DSB repair. However, no significant defects were observed when depleting these ligases independently. Bearing in mind the known redundancy of both ligases, we depleted LIG1 in *LIG3*[−/−] cells, but we only observed a minor defect. These results suggest a model in which TOP1-induced DSB repair might involve the participation of the three repair DNA ligases, and in which LIG4 would participate in a TDP1-independent pathway (Fig. 5f). TDP1 might be a regular factor in processing different 3′-blocked DSBs since the cooperation of TDP1 in cNHEJ has been previously shown in the resolution of 3′-phosphoglycolate termini of DSBs[34], although it has also been associated with the repair of non-blocked DSBs[35].

In agreement with previous studies, our results showed that defects in TDP1, PARP1 and LIG3 increase the formation of replication-independent CPT-induced DSBs[13]. These results suggest that formation of TOP1-induced DSBs is dependent on the accumulation of unrepaired TOP1-induced SSBs (Fig. 5f). TOP1-induced DSB repair defects in *TDP1*[−/−] cells, indicated that TDP1 is essential for the repair of these breaks. Altogether these results suggest that TDP1 is acting at both levels, TOP1-induced SSB repair (and thus preventing the formation of DSBs), and TOP1-induced DSB repair, debulking TOP1 adducts (Fig. 5f). Our repair kinetic analyses also revealed that PARP1 inhibition reduces repair capacity in agreement with PARP1 and TDP1 acting together in TOP1-induced DSB repair[36]. Contrary, PARP1 deficiency did not. The multifaceted role of PARP1 and the dominant negative role of PARP inhibitors is the reason we ascribe this disparity. PARP1-mediated signalling of TOP1-induced SSBs accelerates the recruitment of SSB repair factors and facilitates TOP1cc recognition by TDP1[25]. On the other hand, PARylation can block the proteasomal degradation of abortive TOP1cc, required for TDP1 activity on the TOP1 adduct[37]. Indeed, it has been shown that PARP inhibitors temporally block TOP1cc removal[37]. To our understanding, the fact that PARP inhibitors but not PARP1 deficiency impacts on TOP1-induced DSB repair suggest that accessibility for TOP1 adduct removal by TDP1 is required for an efficient TOP1-induced DSB repair (Fig. 5f). Considering the implication of proteases other than the proteasome in TOP1cc processing, further work is required to completely understand the participation of other factors implicated in abortive TOP1cc signalling and processing during the repair of TOP1-induced DSBs.

Most of our experiments have been performed in quiescent RPE-1 cells to avoid replication-associated DSBs induced by CPT. However, replication-independent TOP1-induced DSBs are not restricted to G0/G1[36] and it is likely that TDP1 would facilitate repair of transcription-associated TOP1-induced DSBs throughout the cell cycle. In agreement, DNAPKcs is hyperactivated in TDP1 deficient cells upon TOP1 poisoning[12], supporting a model in which TDP1 directs the TOP1-

induced DSB repair hierarchy in asynchronous cultures. Inactivating mutations in core factors of cNHEJ suppress genome instability and cell death induced by CPT in the absence of ATM, a key regulator of the DNA damage response[38]. Strikingly, ATM regulates abortive TOP1cc processing and mediates DNAPKcs activation in quiescent cells[10,11], in agreement to the defect in TOP1-induced DSB repair that we observed when we inhibited DNAPKcs. Additionally, ATM is required for efficient repair of DSBs with blocked ends[39]. On the other hand, it has been proposed that cell quiescence could influence DSB repair. For instance, it has been shown that, after ionizing radiation, DNAPK complex uniquely would promote DNA end resection in G0 but not in G1 or G2[40]. It is currently unclear how this complex regulation network would affect TOP1-induced DSB repair.

DSBs associated with transcription are a major source of oncogenic chromosomal translocations[41]. However, the molecular mechanisms behind DSB formation and gene fusion are unclear. Here, we directly asked for the formation of chromosomal rearrangements induced by TOP1 abortive activity. Our results indicate that TDP1-dependent DSB repair suppresses CPT-induced chromosome translocations. Strikingly, these reorganisations were suppressed by RNAPII transcription inhibition. We confirmed this result by directly scoring rearrangements in G0 condensed chromosomes using whole chromosome FISH, discarding that the formation of these rearrangements was the consequence of replication of unrepaired SSBs progressing into S-phase[42]. These results raise the possibility that side effects of TOP1 poison-based chemotherapy would have associated genome instability, similarly to TOP2 poisons, which are a cause of oncogenic chromosomal translocations[43]. In relation to this, TDP1 antagonists are attractive as potential enhancers of TOP1 inhibitors and, to date, many TDP1 inhibitors have been developed. Our data suggest that TDP1 inhibition would potentiate TOP1 poisons not only in proliferative but also in non-proliferative cells. However, our results also anticipate that combination therapy with TDP1 inhibitors and TOP1 poisons might result in genome reorganisations increasing the likelihood of therapy-related diseases.

Several nucleases can participate in the processing of TOP1-induced DSB ends upstream of cNHEJ repair. In the absence of TDP1 activity, an activity capable of removing the bulky residues derived from TOP1cc would also be needed. In agreement, several nucleases such as MRE11, CtIP, XPF, APE2 and MUS81 have been shown to mediate resection in abortive TOP1cc intermediates[4,44]. However, the physiological consequences of these alternative activities are almost unknown. Here, we considered the role of MRE11. Inhibition of MRE11 endonucleolytic activity showed a subtle but significant accumulation of abortive TOP1ccs. Considering the high ratio of SSBs versus DSBs upon CPT treatment, this might indicate the preferential MRE11 activity in DSBs. Notably, our study shows that both depletion and endonuclease inhibition of MRE11 suppressed chromosomal rearrangements, indicating that nucleolytic processing of TOP1-induced DSBs

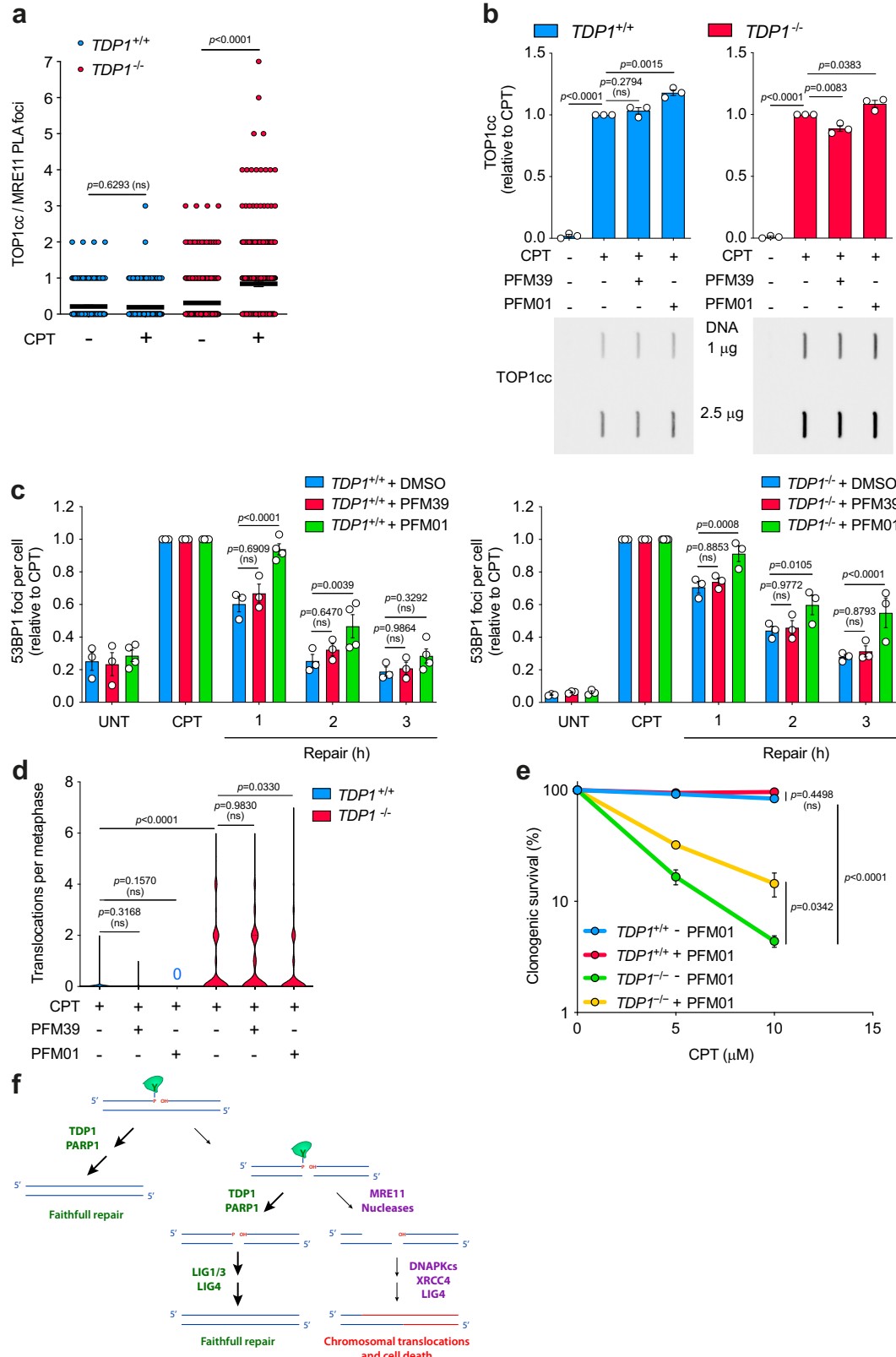

can lead to genome instability (Fig. 5f). A similar scenario has been described in *Saccharomyces cerevisiae*, where replication-independent DSBs are processed by Tdp1 and Wss1 (ortholog of human protease SPRTN). Further processing by Mre11 can promote genomic deletions via cNHEJ[45].

A significant result of this study is the strong cytotoxicity induced by replication-independent abortive TOP1 activity in *TDP1⁻/⁻* cells.

Notably, DRB suppressed CPT sensitivity both in *TDP1⁻/⁻* and wild-type cells demonstrating that transcription associated DSBs can result in cell death. Strikingly, MRE11 inhibition partially rescued sensitivity to CPT of *TDP1⁻/⁻*, evidencing the threat to cell survival of unfaithfully repaired abortive TOP1ccs. Since SSB repair defects are a source of cell death in the brain[5], these results provide insights on the molecular bases of neurodegenerative diseases such as SCAN1 and ATLD

**Fig. 5 | Role of MRE11 in TOP1-induced DSBs repair. a** PLA assay showing TOP1cc-MRE11 proximity in serum-starved *TDP1*⁺/⁺ and *TDP1*⁻/⁻ RPE-1 cells treated with CPT (12.5 μM) for 30 min. From left to right: *n* = 182, *n* = 188, *n* = 286 and *n* = 247 cells over at least 2 independent experiments. **b** Analysis of TOP1 cleavage-complexes (TOP1cc) by ICE assay. Serum-starved *TDP1*⁺/⁺ and *TDP1*⁻/⁻ cells were treated with CPT (25 μM) for 1 h. Where indicated, cells were pre-incubated with MRE11 inhibitors PFM39 (25 μM) or PFM01 (10 μM) for 30 min prior to CPT treatment. *n* = 3 independent experiments. Representative plots of TOP1ccs are shown. Total amount of DNA is indicated. **c** 53BP1 foci in serum-starved *TDP1*⁺/⁺ and *TDP1*⁻/⁻ cells after 1 h treatment with 12.5 μM CPT, and during repair in drug-free medium. Where indicated, cells were pre-incubated with PFM39 (25 μM) or PFM01 (10 μM) for 30 min prior to, during CPT treatment, and during repair. *n* ≥ 3 independent experiments. **d** Translocation frequencies were quantified in serum-starved *TDP1*⁺/⁺ and *TDP1*⁻/⁻ cells in metaphase spreads prepared 48 h after CPT treatment (25 μM)

for 2 h followed by 6 h repair in drug-free medium. Where indicated, cells were pre-treated with PFM39 (25 μM) or PFM01 (10 μM) for 30 min prior to, during, and 6 h after CPT treatment. From left to right: *n* = 201, *n* = 201, *n* = 201, *n* = 217, *n* = 208 and *n* = 367 cells over at least 2 independent experiments. **e** Clonogenic survival of serum-starved *TDP1*⁺/⁺ and *TDP1*⁻/⁻ RPE-1 cells treated with CPT for 2 h, and after 6 h repair in drug-free media. Where indicated, cells were pre-treated with PFM01 (10 μM) for 30 min prior to CPT treatment. After repair, cells were collected and re-cultured in serum containing media. *n* = 3 independent experiments. **f** Model depicting the influence of TDP1 in TOP1-induced DSB repair. UNT untreated. Data were represented as mean ± SEM. Statistical significance was determined by two-tailed unpaired *t*-test for **a**, **b** and **d** and by two-way ANOVA followed by Sidak's multiple comparisons test for **c** and **e**. ns non-significance. Source data are provided as a Source Data file.

syndromes, which are caused by TDP1 and MRE11 deficiencies, respectively.

Altogether, we suggest that the removal of abortive TOP1ccs by TDP1 would initiate a conservative TOP1-induced DSB repair pathway while nucleases such as MRE11 and subsequent cNHEJ would promote genome instability and cell death (Fig. 5f). It should be noted, however, that replication-independent CPT-induced DSBs are heterogeneous. Different pathways may coexist in the cell, depending on the genesis and the nature of DNA end at break site, or its local genomic context. These pathways would likely require different enzymatic activities. Among them, TDP1 phosphodiesterase activity would release the remaining TOP1cc from DNA end. Notably, in the case of MRE11, we detected an accumulation of TOP1ccs and a deficient TOP1-induced DSB repair both in wild-type and *TDP1*⁻/⁻ cells. These results could be reflecting this heterogeneity but also the insufficient capacity of the TDP1-associated route to deal with the high number of CPT-induced DSBs. This would explain the absence of a robust repair defect observed in *LIG4*⁻/⁻ cells and the subtle but significant induction of DNAPKcs-S2056 phosphorylation in TDP1-proficient cells[12]. Considering the strong effects on genome instability and cell survival of CPT in *TDP1*⁻/⁻ cells we suggest the latter as the most likely explanation. Additionally, some studies have shown that tyrosyl-DNA phosphodiesterase 2 (TDP2), a TOP2cc debulking enzyme, can process TOP1 adducts in vitro, and that TDP2 deficiency increases hypersensitivity to CPT in TDP1-deficient cells[46,47]. These results suggest that TDP2 might be able to debulk TOP1-induced DSBs similarly to TDP1. As discussed above, considering the strong effects on genome instability and cell survival of CPT in *TDP1*⁻/⁻ cells, it remains to be proven to which extent TDP2 could back up TDP1 in TOP1-induced DSB repair.

In summary, these data highlight both the threat posed by abortive TOP1 activity to genome stability during transcription and the importance of TDP1-dependent DSB repair in suppressing TOP1-induced cell death and genome instability. This is a significant discovery since, to our knowledge, replication-independent TOP1-induced DSBs have not previously been associated to genome reorganisations in human cells. The repercussions of this for TOP1 poison-based cancer therapy remains to be revealed.

## Methods

### Cell lines and culture conditions
hTERT RPE-1 cells (originally purchased from ATCC, CRL-4000) were propagated in DMEM/F12 medium supplemented with 10% fetal bovine serum (FBS) and with 1% penicillin and streptomycin. For serum starvation, cells were grown until confluency, washed twice with serum-free media, and then cultured in 0% FBS for 3–6 days. HeLa cells (originally purchased from ATCC, CRL-CCL-2) were propagated in DMEM high glucose medium supplemented with 10% FBS and with 1% penicillin and streptomycin. RPE-1 control and *LIG4*⁻/⁻ cells were a generous gift from Professor SP Jackson's Laboratory[38]. XRCC4-V5 expressing RPE-1 cells were a generous gift from F. Cortés-Ledesma's Laboratory (unpublished).

For CRISPR-Cas9-mediated gene targeting TDP1 and POLQ sgRNAs were cloned into the vector #41824 (AddGene) and cotransfected with hCas9 expressed from plasmid #41815 (AddGene). Transfected cells were enriched by selection in 0.5 mg/ml G418 for 5 days prior to isolation of single clones. Screening for loss of TDP1 expression was achieved by western blotting, and for POLQ, by identifying nonsense mutations by PCR. For sgRNA-mediated stable depletion of TDP1, LIG3 and LIG1, cells were infected with lentiviral particles generated using the vector #52961 (AddGene) and selected with 20ug/ml puromycin for 24–48 h. Single clones were screened for loss of TDP1, LIG3 or LIG1 expression by western blotting. Target sequences used in sgRNAs are listed in Table S1.

All cell lines were grown at 37 °C, 5% CO₂ and were regularly tested for mycoplasma contamination. All cell lines tested negative for mycoplasma contamination.

### siRNA transfection
Dharmacon ON-TARGETplus NON-TARGETTING (D-001810-10-05) and MRE11 (L-009271-00-0005), LIG1 (GGCAUGAUCCUGAAGCAGA) and Control (Luciferase CGUACGCGGAAUACUUCGA) siRNA were transfected for 24 h using RNAiMAX (Invitrogen) according to manufacturer's instructions. Then, cells were serum-starved to induce quiescence for 72 h before been treated as indicated.

### Western blotting
Protein extracts were obtained by lysing cell pellets at 100 °C for 10 min in 2x protein buffer (125 mM Tris, pH 6.8, 4% SDS, 0.02% bromophenol blue, 20% glycerol, 200 mM DTT). Extracts were then sonicated in a Bioruptor (Diagenode) for 1 min at high intensity. Primary antibodies were blocked in Tris buffered saline buffer, 0.1% Tween20, 5% BSA and employed as follows: LIG1 (Santa Cruz B, sc-751) 1:1000, LIG3 (GeneTex, GTX70143) 1:1000, LIG4 (Santa Cruz B, sc-271299) 1:100, MRE11 (Novus Biologicals, NB100-142) 1:5000, γH2AX (Millipore, 05-636) 1:1000, TDP1 (Santa Cruz B, sc-365674) 1:250, Vinculin (Santa Cruz B, sc-25336) 1:1000, PARP1 (Thermo Fisher Scientific, 436400) 1:1000 and V5-tag (Abcam, ab15828) 1:2000. Vinculin was used as a loading control. Secondary antibodies (1:5000 dilution in Tris buffered saline buffer 0.1% Tween20 5% BSA): HRP-bovine anti-goat IgG (H + L), HRP-goat anti-mouse IgG (H + L) and HRP-goat anti-rabbit IgG (H + L) (Jackson ImmunoResearch 805-035-180, 115-035-146 and 115-035-144 respectively).

Chemiluminescence data was collected on a ChemiDoc imaging system and analysed in Image Lab 6.0.0 (BIO-RAD). Molecular weight reference is in KDa. Uncropped blots (including molecular weight markers) are provided in Source Data File.

### Immunofluorescence and FISH
For immunofluorescence (IF), cells were grown on coverslips for 2 days (for cycling cultures) or 4–7 days (for serum-starved and confluency-arrested cell cultures) and then treated as indicated. Cells were fixed

(10 min in PBS–4% paraformaldehyde), permeabilized (5 min in PBS–0.2% Triton X-100), blocked (30 min in PBS–5% BSA), and incubated with the indicated primary antibodies for 1–3 h or o/n in PBS–1% BSA. Cells were then washed (3 × 5 min in PBS–0.1% Tween20), incubated for 30 min with the corresponding AlexaFluor-conjugated secondary antibody (1:1000 dilution in PBS–1% BSA) and washed again as described above. Finally, cells were counterstained with DAPI (Sigma, D9542) and mounted in antifade mounting medium for fluorescence (Vectashield, Vector Labs, H-1000). Primary antibodies: anti-γH2AX (Millipore, 05-636) 1:1000 and anti-53BP1 (Novus Biologicals, NB100-904) 1:2500. Secondary antibodies: Alexa Fluor 488-goat anti-mouse IgG (H + L), Alexa Fluor 488-goat anti-rabbit IgG (H + L), Alexa Fluor 546-goat anti-mouse IgG (H + L), Alexa Fluor 546-goat anti-rabbit IgG (H + L) (ThermoFisher Scientific A11001, A11008, A11003, and A11010, respectively). Whole chromosome FISH was performed according to manufacturer's protocol (MetaSystems probes, Whole Chromosome Paint, 739D-0308-050-FI & 739D-0311-050-OR). Click chemistry reaction was performed before DAPI staining by incubating (30 min at room temperature) with 1 mM AlexaFluor-conjugated azide (Invitrogen) in reaction cocktail (100 mM Tris–HCl pH 8.5, 1 mM CuSO4, 100 mM ascorbic acid/ (+)-Sodium L-ascorbate).

Fluorescence intensity of nuclear EU or PAR was obtained using ImageJ 1.52d. DAPI signal was used to delimit the nucleus, and the intercellular background was subtracted.

## 53BP1 repair kinetics
53BP1 foci were scored manually (double blind) in untreated conditions, after treatment with drugs, and during repair in drug-free medium. 53BP1 foci were manually counted (double-blind) in 20–40 cells per data point per independent experiment. Values are shown as the average of 53BP1 foci per cell relative to treatment. Non-normalized 53BP1 repair kinetics are included in Fig. S8.

## Metaphase spreads
For metaphase spreads, cells were incubated with demecolcine (Sigma) at 0.2 mg/ml for 6–20 h and then harvested. Cells were collected using standard cytogenetic techniques, subject to hypotonic shock for 1 h at 37 °C in 0.03 M sodium citrate and fixed in 3:1 methanol:acetic acid solution. Fixed cells were dropped onto acetic acid-humidified slides before dehydration and FISH.

## Chromosomal translocations
Translocation frequencies were calculated as translocations per metaphase in chromosomes 8 and 11, scored manually (double-blind) and plotted together.

## Premature chromosome condensation (PCC)
For premature chromosome condensation (PCC), G0/G1 RPE-1 cells were fused to HeLa cells synchronised in metaphase. To do this, demecolcine was added to cycling HeLa cells. Mitotic HeLa cells were collected by mitotic shake-off. Independently, serum-starved RPE-1 cells were treated as indicated and collected by trypsinization. HeLa and RPE-1 cells were mixed in a 1:5 ratio. Cell mixture was then spined and resuspended in 50% (W/V) polyethylene glycol (PEG) 1500 (Sigma) prepared in serum-free media. Cell mixture was spined and the pellet was resuspended in serum-free media with demecolcine and incubated at 37 °C for 1 h. After that, cell mixture was washed with PBS, and hypotonic shock and cell fixation were performed as described for metaphase spreads.

## Clonogenic survival assays
For asynchronous cells, 300 cells were split in 60 mm dishes. After 6 h cells were treated as indicated, washed with PBS and growth in fresh new media for 10 days. For quiescent cells, G0/G1 cells were treated as indicated, then washed, trypsinized and counted. A total of 400 cells were re-cultured in serum containing media and growth for 8–10 days. In all cases, cells were fixed and stained in PBS-70% ethanol/1% methylene blue. Colonies were counted manually (double blind). The surviving fraction at each dose was calculated by dividing the average number of colonies in treated dishes by the average number in untreated dishes. In all biological replicates cells were split in duplicate for each experimental condition.

## Cell cycle analysis
Cells were incubated with 10 μM BrdU (Sigma, B5002) for 15 min. Cells were washed twice with PBS and fixed with 70% ethanol overnight. DNA was denatured with 2 N HCl/Triton X-100. Cells were incubated with anti-BrdU (Santa Cruz, sc-32323) at 1:1000 overnight at 4 °C. After that, AlexaFluor-conjugated secondary antibody (Invitrogen) was added at 1:1000 during 1 h. Finally, before flow cytometry, cells were incubated with 100 mg/ml PI and 100 mg/ml RNAse A for 30 min. Data was collected in a BD FACSCanto II flow cytometer and analysed in BD FACSDiva Software v9.0.

## ICE assay
ICE assay was performed as previously described[27] with minor modifications. Briefly, a total of $2 \times 10^6$ cells were treated as indicated and lysed with 3 ml 1% Sarkosyl. The lysate was passed through a 25G 5/8" gauge ten times. 2 ml of CsCl solution (1.5 g/ml) was layered into an ultracentrifugation tube. The volume of lysate was layered on top of the gradient. Samples were centrifuged in a NVt90 rotor at 25 °C, 121,900 g for 20 h. Pellet was resuspended in TE 1× and DNA concentration was measured in a Nanodrop. 1 μg and 2.5 μg of DNA were loaded in nitrocellulose membrane preincubated in 25 mM NaPO4 pH 6.5 for 15 min using a slot-blot apparatus. Abortive TOP1ccs were detected by TOP1cc antibody[28] (Millipore, MABE1084) 1:250.

## Comet assay
Cells were grown until confluency, then serum-starved for 3 days, collected and treated in suspension in 0% FBS medium as indicated. After treatment, cells were washed once and resuspended in 0.5 ml ice-cold PBS. Alkaline comet assay was performed as previously described[48]. Briefly, cells were mixed with an equal volume of 1.2% low melting point agarose (Lonza, 50080) in PBS (at 42 °C). Cell suspension was immediately layered onto pre-chilled frosted glass slides pre-coated with 0.6% agarose and maintained in the dark at 4 °C until agarose set. Slides were then immersed in pre-chilled alkaline lysis buffer (2.5 M NaCl, 10 mM Tris–HCl, 100 mM EDTA, 10 mM Tris-Cl, 1% v/v DMSO, 1% v/v Triton X-100, pH 10) at 4 °C for 1 h and then washed three times with pre-chilled distilled water. Samples were incubated in pre-chilled alkaline electrophoresis buffer (1 mM EDTA, 50 mM NaOH, 1% v/v DMSO) for 45 min to facilitate DNA denaturation, prior to electrophoresis at 0.6 V/cm for 25 min at 4 °C. Following electrophoresis, slides were neutralized in 0.4 M Tris-HCl pH 7 for 1 h and then stained with 1x SYBR green (Sigma S9430) and 37 μl of antifade mounting medium for fluorescence for 10 minutes in PBS. For neutral comet assay[24], after incubation in lysis buffer (2.5 M NaCl, 10 mM Tris–HCl, 100 mM EDTA, 1% N-laurosylsarcosine, 10% v/v DMSO, 0.5% v/v Triton X-100, pH 9.5) at 4 °C for 1 h cells were washed three times and incubated with pre-chilled electrophoresis buffer (300 mM sodium acetate, 100 mM Tris-HCl, bring up to pH 8.3) for 1 h at 4 °C. Electrophoresis was run at 0.5 V/cm for 1 h. Following electrophoresis, slides were incubated in 0.4 M Tris-HCl pH 7 for 1 h before SYBR green staining.

Slides were visualized by using a fluorescence microscope (Olympus BX-61). Values are shown as the quantification of comet tail moments. In all cases experiments were analysed by CometScore Pro software.

## Proximity ligation assay (PLA)

Duolink PLA assay (Sigma, DUO92101) was performed conducting the manufacturer's instructions. Briefly, cells were grown on coverslips until confluency, then serum-starved for 3 days and treated as indicated. Cells were fixed (10 min in PBS–4% paraformaldehyde), permeabilized (5 min in PBS–0.2% Triton X-100), blocked (30 min in Duolink blocking solution), and incubated with the required primary antibodies for 90 min in Duolink antibody diluent at 37 °C. After that, incubation with secondary antibodies conjugated with oligonucleotides was performed, followed by ligation and amplification. Finally, coverslips were mounted with Duolink in situ mounting medium with DAPI. Primary antibodies were used as indicated for IF, ICE and western blotting.

## Statistical analysis

Statistical analysis is included in figure legends. In all cases, comparison tests were performed using GraphPad Prism version 8.2.1 for macOS.

## Reporting summary

Further information on research design is available in the Nature Portfolio Reporting Summary linked to this article.

## Data availability

The data supporting the findings of this study are available from the corresponding authors upon request. Source data are provided with this paper. Uncropped blots including 1c, 2a, 2h, 3a, 3b, 3c, 3d, 3e, 5b, S4, S5a, S5b and S7a are provided with this paper. Source data are provided with this paper.

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

## Acknowledgements
We thank Sebastián Chavez, Jesús de la Cruz and Iván V. Rosado laboratories for discussion, and Felipe Cortés-Ledesma, William Gittens and Jose F. Ruiz for critical reading of the manuscript. We also thank Felipe Cortés-Ledesma for providing unpublished RPE-1 XRCC4-V5 cell line. We thank Margarita Sabio-Bonilla for technical assistance cloning LIG1 and TDP1 gRNAs into plentiV2. We also thank Fernando Gómez-Aranda for his support. This publication is part of the project R + D + i PID2019-105212GB-I00 funded by MCIN/AEI/10.13039/501100011033 to F.G-H. We also acknowledge the Andalusian Regional Government [P20_00561, BIO-271 to F.G-H.]; D.R-C. is recipient of a predoctoral fellowship from University of Sevilla [VI PPIT-US]. Funding for open access charge: PID2019-105212GB-I00 funded by MCIN/AEI/10.13039/501100011033 (to F.G-H.).

## Author contributions
D.R-C. and F.G-H. designed and performed the experiments, interpreted the results, and wrote the manuscript. F.G-H. conceived and directed the study.

## Competing interests
The authors declare no competing interests.
