## [Peer Review File · Nature Communications]

REVIEWER COMMENTS

Reviewer #1 (Remarks to the Author):

In the present study, Rubio-Conteras and Gómez-Herreros investigated the repair pathway of transcription-dependent, but replication-independent, CPT-induced DSBs. The authors found that TDP1 played a significant role for repairing CPT-induced DSBs in the G1 phase, and that LIG1/3/4 also contributes to DSB repair. Depletion of TDP1 led to an increase in aberrant chromosome formation, which is likely associated with the incidence of cell death in the G1 phase after CPT treatment. Furthermore, the authors showed that translocation and cell death required MRE11 endonuclease activity particularly when TDP1 was depleted. Overall, the manuscript is well-written and all the experiments are well-designed. The proposed model is plausible. However, a critical problem in this study is the lack of a molecular mechanism. Most of the factors analyzed in this study have been previously suggested or readily predicted. In addition, the conclusion is supported only by genetic analyses using basic molecular biology techniques such as foci analysis (IF and PLA), chromosomal analysis, and a cell viability assay. Thus, the molecular mechanism underlying the functional interplay between Top1-cc, TDP1, LIG1/3/4, and MRE11 remains unclear. I believe that further analyses to elucidate the mechanism are critical to publish this work in a high impact factor journal. Nevertheless, because the quality of the manuscript is sufficient, I would suggest that a specialized journal for DNA repair is suitable for this study.

1. As previously known, 53BP1 foci in G1 after CPT is transcription-dependent (PMID: 20304914). The authors should appreciate and acknowledge this finding.
2. In Figure 5B, why did PFM01 show such a minor effect in TDP1^{-/-} cells although the impact on translocation and cell viability is high? How about the impact on DSB repair by PFM01? In addition, are other DNA nucleases, e.g. CtIP, involved in the removal of TOPcc in TDP1^{-/-} cells?
3. The mechanisms of the requirement (or recruitment) of LIG4 and MRE11 are unclear. How does MRE11 endonuclease activity contribute to the repair? Is it just to remove the dirty DSB end? Or is MRE11 also involved in the step of resection, which may require Artemis etc. after Top1-cc removal?
4. 53BP1 is a marker of DSBs when X-rays or gamma-rays are used. If the LIG4-dependency is low, it is very difficult to judge whether DSBs are truly formed after CPT in the G1 phase? The authors should verify the presence of DSBs by a marker-independent method (e.g., END-seq or a similar sequence-based method) and should check whether DSBs are induced at transcription-active loci.

Reviewer #2 (Remarks to the Author):

In this manuscript, the authors investigated the roles of TDP1 in topoisomerase I poison induced DNA damage repair. Using confluent cells and serum starvation, the authors arrested RPE-1 cells at G0/G1 phase and studied the repair of TOP1 poison/CPT-induced DNA lesions. Similar to early studies, the authors showed that the repair of transcription-associated CPT induced DNA damage requires TDP1 and at least in part cNHEJ pathway. Additionally, they showed that in the absence of TDP1, cells could engage error-prone repair pathways, which led to chromosomal translocation and cytotoxicity.

In general, the data presented are convincing. However, there is no strong evidence to support the TDP1-dependent DSB repair pathway as speculated by the authors. It is likely that CPT would induce more SSBs in TDP1 KO G0/G1 cells, which are processed by nucleases and others to generate additional DSBs that need to be repaired by cNHEJ and other repair pathways. Indeed, the authors showed that CPT-treatment led to increased SSBs as measured by alkaline comet assay in Fig. 2B and also increased DSBs as measured by 53BP1 foci in Fig. 2D. Therefore, TDP1 KO cells have increased DSBs following CPT treatment, but not that TDP1 is involved in DSB repair.

Specific comments:

1. The increased DSBs were not apparent in Fig. 2E-G or Fig. 3, which may be due to the saturation of their 53BP1 assay system or that they used the relative to CPT as the Y axis. DSB repair is not a linear process and therefore should not be presented this way. In addition, in line with the alkaline comet assay for SSBs, the authors should include neutral comet assay for DSBs.
2. Early studies using double KO cells suggest that DSB repair mainly depend on cNHEJ and POLQ-dependent pathways. It is unlikely that POLQ inhibitor would be as effective as POLQ KO cells. The authors should use DKO cells, but not POLQ inhibitor, to rule out the potential role of POLQ in CPT-induced DSB repair. In addition, the authors should also use these single and DKO cells to investigate the role of POLQ in chromosomal translocation.
3. The authors used PARPi and other inhibitors to draw conclusions. However, these inhibitors may not totally inhibit the proposed enzymatic activity or they may have dominant negative effects such as PARP trapping. Whenever possible, the authors should include genetic experiments, i.e. DKO cells, to support their conclusions.

Reviewer #3 (Remarks to the Author):

Conteres and Gomez-Herreros have explored how Top1 induced double strand breaks are repaired in quiescent cells. The standard model of camptothecin action, as originally proposed by Liu and co-workers, is that replication fork collisions with Top1 covalent complexes lead to the production of single-ended double strand breaks. More recent results have demonstrated that there are also replication independent induced Top1 induced breaks. The present work bears on how replication independent breaks are repaired. The key finding is that in the absence of Tdp1, the repair of Top1 induced breaks can frequently lead to translocations mediated by canonical non-homologous end-joining. I think this result will be of interest to the topoisomerase community and to cancer chemotherapy investigators. Most of my comments relate to the interpretation of their results, and some suggestion that the limitations and implications of their work are incompletely described.

Replication independent breaks have been studied by others, and that work is appropriately cited here. The first point to highlight is what is the mechanism of replication induced breaks? Work from Junjie Chen showed that MUS81 plays a role in some Top1 induced breaks, but not all breaks depend on MUS81. The authors need to carefully rephrase lines 78-80 regarding the genesis of Top1 mediated double strand breaks to clearly indicate that the breaks arise from multiple sources and to indicate that the breaks studied in this paper arise from heterogeneous sources. This is especially important when we come to the question of how the breaks are repaired, since much of the authors text suggest a homogenous population of dsbs, which need not be the case.

A key question that comes from this paper is how does Tdp1 participate in the repair of Top1 induced breaks? The authors never provide a description of the Tdp1 dependent pathway.

The authors note that the repair by the Tdp1 dependent pathway does not lead to observable translocations, and they call this repair error free. The data do not support the conclusion that the repair is error free. The lack of translocations does not mean the repair is error free, and the authors need to be more precise in their description.

The authors are imprecise when they note that the recruitment of TDP1 depends entirely on PARP. There are clearly two pathways; a PARP dependent pathway and a SUMO dependent pathway (as described by Das et al., 2014, cited in this paper but not clearly incorporated into the discussion.

The authors show, as others have done, that loss of Tdp1 leads to elevated levels of intact Top1 complexes (Figure 5b). Since proteolysis seems to be important for TDP1 processing, why isn't the lesion seen in TDP1- cells proteolyzed Top1? The authors should examine whether the elevated levels of trapped Top1 are actually intact Top1 or whether it is partially proteolyzed but recognized by the antibody that they use.

There has been significant interest in developing TDP1 inhibitors to enhance the cytotoxicity of camptothecins and other Top1 inhibitors. The authors' results suggest that this strategy might be perilous, since it leads to Top1 mediated translocations. This possibility should be discussed.

Overall, the most important concerns raised here are:

- (1) How does Tdp1 lead to repair of Top1 induced double strand breaks.
- (2) If the authors results are correct, what does this suggest about the potential of Tdp1 inhibitors?

Minor points and corrections

Lines 169-170, novobiocin is a specific and potent inhibitor of POLQ; novobiocin may be potent but it is certainly not specific. Novobiocin targets bacterial topoisomerases and has weak inhibitory activity against eukaryotic type II topoisomerases. Novobiocin results led to significant confusion regarding the roles of topoisomerases in repair. I have no problem using this agent in a supporting experiment, but it is incorrect to call novobiocin specific.

Reviewer #1 (Remarks to the Author):

In the present study, Rubio-Conteras and Gómez-Herreros investigated the repair pathway of transcription-dependent, but replication-independent, CPT-induced DSBs. The authors found that TDP1 played a significant role for repairing CPT-induced DSBs in the G1 phase, and that LIG1/3/4 also contributes to DSB repair. Depletion of TDP1 led to an increase in aberrant chromosome formation, which is likely associated with the incidence of cell death in the G1 phase after CPT treatment. Furthermore, the authors showed that translocation and cell death required MRE11 endonuclease activity particularly when TDP1 was depleted. Overall, the manuscript is well-written and all the experiments are well-designed. The proposed model is plausible. However, a critical problem in this study is the lack of a molecular mechanism. Most of the factors analyzed in this study have been previously suggested or readily predicted. In addition, the conclusion is supported only by genetic analyses using basic molecular biology techniques such as foci analysis (IF and PLA), chromosomal analysis, and a cell viability assay. Thus, the molecular mechanism underlying the functional interplay between Top1-cc, TDP1, LIG1/3/4, and MRE11 remains unclear. I believe that further analyses to elucidate the mechanism are critical to publish this work in a high impact factor journal. Nevertheless, because the quality of the manuscript is sufficient, I would suggest that a specialized journal for DNA repair is suitable for this study.

We thank the critical analysis made by Reviewer1. We think that addressing his/her major points and with the new data incorporated to the manuscript this work has contributed to answer mechanistic questions about TOP1-induced DSB repair and the formation of replication-independent CPT-induced chromosomal translocations. Please find our response below.

1. As previously known, 53BP1 foci in G1 after CPT is transcription-dependent (PMID: 20304914). The authors should appreciate and acknowledge this finding.

Thank you very much for indicating that we had missed this previous contribution. This reference has now been added as part of discussion (379-381).

2. In Figure 5B, why did PFM01 show such a minor effect in TDP1^{-/-} cells although the impact on translocation and cell viability is high? How about the impact on DSB repair by PFM01? In addition, are other DNA nucleases, e.g. CtIP, involved in the removal of TOPcc in TDP1^{-/-} cells?

Thank you very much for these comments. It has guided us to provide more mechanistic data regarding the role of MRE11 in TOP1-induced DSB repair.

To our understanding, there can be several reasons for this difference. First, as shown by others, TDP1^{-/-} cells accumulate higher levels of abortive TOP1ccs than wild-type cells (Fig S4 & 5B). Thus, in TDP1^{-/-} cells, there is a smaller window to see variations than in wild-type cells considering the sensitivity of ICE. Second, accumulation of TOP1cc, the formation of chromosomal translocations and CPT sensitivity are very different readouts of TOP1 abortive activity and there may not be a direct proportionality between these events. Third, we discuss in the manuscript that the high ratio SSB/DSB upon CPT treatment might also explain this minor but significant difference in TOP1cc. However, we find these differences compatible with our conclusions.

On the other hand, we found the suggestion of testing the role of endonuclease activity of MRE11 in TOP1-induced DSB repair a very relevant request. In this reviewed version we have included these experiments. We have performed repair kinetics in WT and TDP1^{-/-} cells exposed to PFM01. Inhibition of MRE11 endonuclease activity resulted in a significant defect in TOP1-induced DSB repair suggesting that MRE11 endonuclease activity is required for this repair process in a TDP1-independent pathway (New Fig 5C). Notably, we not only tested PFM01 but PFM39 as well, an MRE11 exonuclease activity inhibitor. Please see comments on PFM39 below as a response to major point 3.

Finally, regarding the participation of other nucleases, we assume that this is a very likely possibility as we discuss it in the manuscript (460-476). However, we decided to concentrate our efforts on MRE11 due to previous evidence of MRE11 activity on abortive TOP1cc. Nevertheless, to answer Reviewer3 suggestion we depleted CtIP and score 53BP1 foci. We did not see a significant accumulation of DSBs (Fig. R1). These results are similar to those described by Cristini et al Cell Reports, PMID: 31533039.

Figure R1. Average 53BP1 foci in mock-depleted (siLuc) or CtIP-depleted (siCtIP) serum-starved RPE-1 cells treated with CPT (25 μ M) for 1 h. Data are the mean (\pm s.e.m.) of three independent experiments. Statistical significance was determined by t-test (**P < 0.001; NS, not significant)

3. The mechanisms of the requirement (or recruitment) of LIG4 and MRE11 are unclear. How does MRE11 endonuclease activity contribute to the repair? Is it just to remove the dirty DSB end? Or is MRE11 also involved in the step of resection, which may require Artemis etc. after Top1-cc removal?

We thank Reviewer1 for pointing this mechanistical question. Based on the concern from Reviewer1 we have tested accumulation of TOP1ccs by ICE, DSB repair kinetics and chromosomal translocations using PFM39, an MRE11 exonuclease inhibitor. Notably, contrary to the significant effect of PFM01 we did not detect the accumulation of TOP1ccs using PFM39. Strikingly, inhibition of the exonucleolytic activity of MRE11 did not affect the repair of TOP1-induced DSBs and did not suppress CPT-induced chromosomal translocations. These results suggest that TOP1cc removal is a key role of MRE11. PFM39 prevents resection inhibiting homologous recombination (HR) (Fig. R2, Shibata et al. Mol Cell 2014 PMID: 24316220). Therefore these results are against a possible implication of MRE11-mediated resection in TOP1-induced DSB repair in our studies. Please find these results included in new Fig 5B, 5C, 5D. Our results strongly suggest that removal of TOP1, TOP1-DSB repair and chromosomal translocation formation are mediated by the endonucleolytic activity of MRE11.

Figure R2. Sister chromatid exchange (SCEs) of RPE-1 treated with 5 μ M etoposide for 30 min. Where indicated cells were pre-incubated with RAD51 inhibitor RI-1 (10 μ M) or PFM39 (25 μ M) for 60 min prior to, during and after etoposide treatment. Cells were cultured in drug-free medium for 24 h previous to metaphase spread preparation. Mean (\pm s.e.m.) of SCE events per chromosome per metaphase. Statistical significance was determined by T-test (**P < 0.01, ***P < 0.001, NS, not significant)

4. 53BP1 is a marker of DSBs when X-rays or gamma-rays are used. If the LIG4-dependency is low, it is very difficult to judge whether DSBs are truly formed after CPT in the G1 phase? The authors should verify the presence of DSBs by a marker-independent method (e.g., END-seq or a similar sequence-based method) and should check whether DSBs are induced at transcription-active loci.

Thank you for this comment. The formation of replication independent DSBs upon TOP1-poisoning has been shown by other laboratories in multiple cellular models. Additionally, in this manuscript, the detection of chromosomal translocations by PCC-FISH is a direct evidence that CPT induces DSBs in G0/G1 (Fig 4D). Nevertheless, to answer to your concern and to directly measure DSBs by a marker-independent method we conducted neutral comet assays. As Reviewer1 is probably aware neutral comets are much less sensitive than alkaline comets and are mainly useful to measure very high number of DSBs. Indeed, some authors have suggested that sensitivity of this technique starts from 40-50 DSBs (Wojewódzka et al, Mut Res 2002, PMID: 12063063). It is important to consider that working on non-cycling cells strongly reduces CPT-induced comet tails due to the low number of DSBs. That is the reason we found 53BP1 and gH2AX foci much more sensitive and appropriate to evaluate rapid DSB repair. Nevertheless, based on the concern from Reviewer1 we have analysed DSB formation and repair by neutral comets in wild-type and TDP1 KO cells (New Fig 2G). We observed a significant increase in comet tails upon CPT treatment in wild-type cells (New Fig 2G). This increase was significantly higher in TDP1^{-/-} cells (New Fig 2G). Additionally, TDP1^{-/-} deficiency resulted in a deficient repair of these breaks (New Fig 2G). We think that these new results support our previous conclusions.

We also thank Reviewer1 suggestion about using a sequence-based method to characterise TOP1-induced DSBs. This is in fact a very interesting point and a running project in the laboratory. However, as Reviewer1 is probably aware (and Reviewer3 has indicated), replication-independent TOP1-induced DSBs are likely a highly heterogeneous population, with asymmetric and dirty DSB ends. Therefore, sequencing TOP1-induced DSBs is a very complex task and requires a fit for purpose technique. We are putting our efforts in adapting different techniques to these particular DSBs in order to generate good libraries and obtain robust maps. However, this is a long-term project in the laboratory and we are facing with several technical difficulties. In summary, although we like the idea of mapping TOP1-induced DSBs, this was not feasible in the time period available to us. We hope the reviewer and editor understand our position.

Reviewer #2 (Remarks to the Author):

In this manuscript, the authors investigated the roles of TDP1 in topoisomerase I poison induced DNA damage repair. Using confluent cells and serum starvation, the authors arrested RPE-1 cells at G0/G1 phase and studied the repair of TOP1 poison/CPT-induced DNA lesions. Similar to early studies, the authors showed that the repair of transcription-associated CPT induced DNA damage requires TDP1 and at least in part cNHEJ pathway. Additionally, they showed that in the absence of TDP1, cells could engage error-prone repair pathways, which led to chromosomal translocation and cytotoxicity.

In general, the data presented are convincing. However, there is no strong evidence to support the TDP1-dependent DSB repair pathway as speculated by the authors. It is likely that CPT would induce more SSBs in TDP1 KO G0/G1 cells, which are processed by nucleases and others to generate additional DSBs that need to be repaired by cNHEJ and other repair pathways. Indeed, the authors showed that CPT-treatment led to increased SSBs as measured by alkaline comet assay in Fig. 2B and also increased DSBs as measured by 53BP1 foci in Fig. 2D. Therefore, TDP1 KO cells have increased DSBs following CPT treatment, but not that TDP1 is involved in DSB

repair.

We acknowledge Reviewer2 constructive comments. This concern is in fact a very important point. Previous studies and our results demonstrate that defects in SSB repair promotes the formation of replication-independent TOP1-induced DSBs. Indeed, defects in any factor involved in this repair (PARP1, TDP1, PNKP, LIG3), some of them downstream of TDP1, promote this accumulation (Cristini et al, Cell reports 2019, PMID: 31533039, among others). In our model (New Fig 5F) we indicate that formation of TOP1-induced DSBs is dependent on the accumulation of SSBs, suggesting that TDP1 is acting at both levels, SSB repair (and thus preventing DSBs), and DSB repair. We think that the suggestion of Reviewer2 regarding PARP1 inhibition has helped us clarifying this point. Please see response to specific points 1 and 3 below.

Specific comments:

1. The increased DSBs were not apparent in Fig. 2E-G or Fig. 3, which may be due to the saturation of their 53BP1 assay system or that they used the relative to CPT as the Y axis. DSB repair is not a linear process and therefore should not be presented this way. In addition, in line with the alkaline comet assay for SSBs, the authors should include neutral comet assay for DSBs.

Thank you very much for this comment since this is a very important point. Our results demonstrate that defects in TOP1-induced SSB repair promotes TOP1-induced DSBs. This is not a novel observation but in agreement with previous reports in other cellular systems. Consequently, TDP1^{-/-} cells accumulate many more DSBs than wild-type cells upon CPT treatment (Fig 2D). Comparing DSB repair kinetics of wild-type and TDP1^{-/-} cells is therefore not simple since there is a big difference in induction. This is the reason why we decided to normalise DSBs using the number of 53BP1 foci upon CPT treatment so that the repair kinetics after drug wash would be comparable. Based on the concern from Reviewer2 we have now included non-normalised kinetics (New Fig. S8) in which we show the average foci number of all replicates of all repair kinetics indicating the background level. In all cases those differences in repair rates observed in normalized kinetics are also apparent when the whole kinetic is compared. However, we believe that the way we presented the data facilitates the visualization of the phenotype for the reader. Several relevant studies have used DSB repair kinetics before and this type of plot is the most common used by most authors. We hope Reviewer 2 finds this decision the most sensible.

On the other hand, as Reviewer2 is probably aware neutral comets are much less sensitive than alkaline comets and are useful to measure very high number of DSBs. Indeed, some authors have suggested that sensitivity of this technique starts from 40-50 DSBs (Wojewódzka et al, Mut Res 2002, PMID: 12063063). It is important to consider that working on non-cycling cells strongly reduce CPT-induced comet tails due to the low number of DSBs. That is the reason we found 53BP1 and gH2AX foci scoring much more sensitive and appropriate to evaluate rapid DSB repair. Nevertheless, based on the concern from Reviewer2 we have analysed DSB formation and repair by neutral comets in wild-type and TDP1 KO cells. We observed a significant increase in comet tails upon CPT treatment in wild-type cells (New Fig 2G). This increase was significantly higher in TDP1^{-/-} cells (New Fig 2G). Additionally, TDP1^{-/-} deficiency resulted in a deficient repair of these breaks (New Fig 2G). We think that these new results support our previous conclusions, and we hope that Reviewer2 finds them appropriate.

2. Early studies using double KO cells suggest that DSB repair mainly depend on cNHEJ and POLQ-dependent pathways. It is unlikely that POLQ inhibitor would be as effective as POLQ KO cells. The authors should use DKO cells, but not POLQ inhibitor, to rule out the potential role of POLQ in CPT-induced DSB repair. In addition, the authors should also use these single and DKO cells to investigate the role of POLQ in chromosomal translocation.

Based on the suggestion from Reviewer2 we tried to generate a LIG4 POLQ DKO. However, we repeatedly failed. By using the same double-sgRNA approach we used to generate TDP1POLQ DKOs (Fig. S3C) we obtained several POLQ KO clones in control cells but not in LIG4 KOs. This

was an unexpected result since previous reports have demonstrated that LIG4 POLQ DKO cells in Nalm-6 cells are not lethal (Saito et al Nat Comms 2017, PMID: 28695890). LIG4 KO cells are very sick so it might just be that this slow growth together with the stringent clonal selection significantly reduces the probability of success. Since this is not the main aim of the study and this was delaying the submission, we used an additional drug-based approach (see below). We hope the reviewer and editor understand our position.

Considering the difficulties generating LIG4 POLQ DKO we decided to repeat repair kinetics and translocations using ART558, a novel, very potent and much more specific inhibitor of POL θ than NVB (Zatreanu et al, Nat Comms, PMID: 34140467). ART558 has been proved to efficiently inhibit POLQ-mediated end Joining (TMEJ) (Zatreanu et al, Nat Comms, PMID: 34140467). Notably, we obtained the same result than using NVB (new Fig 1F), supporting our previous conclusions.

Regarding the role of POLQ in CPT-induced chromosomal translocations, since LIG4^{-/-} cells do not accumulate chromosomal translocations upon CPT treatment (Fig S6C), instead of LIG4 POLQ DKO we checked the formation of chromosomal translocations in TDP1POLQ DKO and inhibiting POLQ in TDP1^{-/-} cells (please see new Fig. S6D). Notably, we did not detect any significant change in chromosomal translocations induced by CPT suggesting that POLQ is not implicated in these rearrangements. We think that these new results support our previous conclusions.

We think that the fact that we do not see a POLQ dependency might be related to quiescence since POLQ implication in DSB repair has been lately associated to S-G2/M (Yu et al Nat Comms 2020, PMID: 33067475 & Belan et al Mol Cell 2022, PMID: 36455556; among others). On the other hand, a very recent publication demonstrated that POLL promotes MMEJ in mammalian cells independently of LIG4/XRCC4 and POLQ (Chandramouly et al NSMB 2023, PMID: 36536104). This make us think that very likely there are other redundant pathways (not necessarily POLL) that could explain remaining TOP1-induced DSB repair in LIG4 + ART558 and LIG4 + NVB repair kinetics and that we may have not considered. To be more accurate in the description of our results we decided to change altNEHJ for TMEJ (160-169). We hope Reviewer2 find these changes adequate to his/her request.

3. The authors used PARPi and other inhibitors to draw conclusions. However, these inhibitors may not totally inhibit the proposed enzymatic activity or they may have dominant negative effects such as PARP trapping. Whenever possible, the authors should include genetic experiments, i.e. DKO cells, to support their conclusions.

Thank you very much for this comment since this is a very important point. We truly think that addressing it has helped us improving the manuscript although it has opened some new questions. We totally agree with Reviewer2 that small molecule inhibitors present many problems and do not always recapitulate lack-of function phenotypes. This is in fact especially notorious with PARP inhibitors (PARPi) that have been shown by many studies to trap PARP1 on the break, provoking a dominant phenotype. However, using PARPi the way we did there was no need to normalise repair kinetics. That is the reason why we added PARPi just after CPT wash so that we could also discard remaining SSBs becoming DSBs during repair (old Fig 2G). Nevertheless, to address the very sensible Reviewer2 suggestion, we have now tested TOP1cc accumulation by ICE, repair kinetics and chromosomal translocations in PARP1^{-/-} cells upon CPT treatment. Notably, these experiments have been very clarifying regarding the participation of TDP1 in TOP1-induced DSB repair and have provided novel mechanistic insights to our model. PARP1^{-/-} cells showed very significant accumulation of DSBs upon CPT addition (similar to those provoked by TDP1^{-/-})(New Fig. 2H). However, PARP deficiency did not delay TOP1-induced DSB repair (New Fig 2I) and did not promote a high number of TOP1-induced chromosomal translocations (New Fig. 4C). Considering that PARP1^{-/-} cells showed a lower accumulation of TOP1cc than TDP1^{-/-} cells and RPE-1 cells treated with PARP inhibitors, to our understanding these results suggest that:

1. there is a proportional formation of DSBs from TOP1-induced SSBs, not necessarily dependent on TOP1 trapping.
2. TOP1-induced DSB repair defects are associated to the removal of TOP1cc adduct.
3. TOP1-induced chromosomal translocations are not only dependent on TOP1-induced DSB but on defects on TOP1cc removal.

We hope Reviewer2 is satisfied with these new results and agrees with our conclusions.

Reviewer #3 (Remarks to the Author):

Conteres and Gomez-Herreros have explored how Top1 induced double strand breaks are repaired in quiescent cells. The standard model of camptothecin action, as originally proposed by Liu and co-workers, is that replication fork collisions with Top1 covalent complexes lead to the production of single-ended double strand breaks. More recent results have demonstrated that there are also replication independent induced Top1 induced breaks. The present work bears on how replication independent breaks are repaired. The key finding is that in the absence of Tdp1, the repair of Top1 induced breaks can frequently lead to translocations mediated by canonical non-homologous end-joining. I think this result will be of interest to the topoisomerase community and to cancer chemotherapy investigators. Most of my comments relate to the interpretation of their results, and some suggestion that the limitations and implications of their work are incompletely described.

We are very grateful to Reviewer3 for the very constructive review of the manuscript he/she has done. We hope that the experiments and text modifications that we have done match his/her requests.

Replication independent breaks have been studied by others, and that work is appropriately cited here. The first point to highlight is what is the mechanism of replication induced breaks? Work from Junjie Chen showed that MUS81 plays a role in some Top1 induced breaks, but not all breaks depend on MUS81. The authors need to carefully rephrase lines 78-80 regarding the genesis of Top1 mediated double strand breaks to clearly indicate that the breaks arise from multiple sources and to indicate that the breaks studied in this paper arise from heterogeneous sources. This is especially important when we come to the question of how the breaks are repaired, since much of the authors text suggest a homogenous population of dsbs, which need not be the case.

We completely agree with Reviewer 3 about the expected heterogeneity of TOP1-induced DSBs and it was our intention to indicate it in the manuscript. Obviously, we did not do it properly. In this new version we have included a more explicit sentence in the introduction (73-75) and extended the discussion about this fact and how it limits our conclusions (490-499).

A key question that comes from this paper is how does Tdp1 participate in the repair of Top1 induced breaks? The authors never provide a description of the Tdp1 dependent pathway.

Thank you for this question. Altogether our data strongly suggest that the removal of abortive TOP1cc by TDP1 is a key step to prevent TOP1-induced chromosomal translocations. We have extended the discussion about the role of TDP1 in the manuscript. Please see response to concern (1) below.

The authors note that the repair by the Tdp1 dependent pathway does not lead to observable translocations, and they call this repair error free. The data do not support the conclusion that the repair is error free. The lack of translocations does not mean the repair is error free, and the authors need to be more precise in their description.

Thank you very much for pointing this inaccuracy. We totally agree with this comment. We expect to have the genome distribution of TOP1-induced DSBs in the future so that we can directly measure the mutagenic consequences of these breaks and how TDP1 and other factors prevent/cause them. Based on the suggestion from Reviewer3 we have corrected the model removing "error-free". We have also removed "error-free repair" from text.

The authors are imprecise when they note that the recruitment of TDP1 depends entirely on PARP. There are clearly two pathways; a PARP dependent pathway and a SUMO dependent pathway (as described by Das et al., 2014, cited in this paper but not clearly incorporated into the discussion.

Thank you very much for marking this deficiency. Please find this sentence corrected in the new version (193). Since several experiments using PARP1^{-/-} cells are now included in the manuscript (New Fig. 2H, 2I, 2J, 4C & S4) we have included an extended discussion on the role of PARP1 and the removal of abortive TOP1cc in TOP1-induced DSB repair (413-424). On the other hand, experiments with PARP1^{-/-} cells have brought several new questions. We hope Reviewer3 finds these results interesting and properly argued.

The authors show, as others have done, that loss of Tdp1 leads to elevated levels of intact Top1 complexes (Figure 5b). Since proteolysis seems to be important for TDP1 processing, why isn't the lesion seen in TDP1^{-/-} cells proteolyzed Top1? The authors should examine whether the elevated levels of trapped Top1 are actually intact Top1 or whether it is partially proteolyzed but recognized by the antibody that they use.

We thank Reviewer 3 for addressing this issue and we think he/she is completely right. To detect TOP1ccs we have used MABE1084 antibody, published by Patel et al (NAR 2016 PMID: 26917015). We had missed this reference that is now included in the new version of the manuscript (621-622). This antibody was raised against a peptide corresponding to the active site of the TOP1 with a phosphorylated Tyr723 residue (716 LGTSKLN(phosphoY)LDPRITV 730). This antibody detects DNA-bound TOP1 (TOP1cc) by IF (Patel et al, NAR 2016 PMID: 26917015). We reasoned that using this antibody we would detect both intact and partially proteolyzed TOPcc. Based on the suggestion from Reviewer3, we have performed ICE assays combining CPT and proteasomal inhibition (MG132). We compared MABE1084 and the C21 monoclonal antibody raised against complete TOP1 (sc-32736). Proteasomal inhibition resulted in increased ICE signal in wild-type cells treated with CPT using both antibodies (Fig. R3, blue bars), suggesting that both antibodies can detect intact TOP1cc. Strikingly, this increase was also observed in TDP1^{-/-} cells with C-21 but not with MABE1084 that showed a decrease (Fig. R3, red bars). These results would suggest that excess TOP1 signal detected by MABE1084 in TDP1^{-/-} cells is, at least partly, partially proteolyzed TOP1. To our understanding these results suggest that using "abortive TOP1cc" to define ICE signal in our manuscript is not inadequate. We wonder whether Referee3 would agree with our point of view.

Figure R3. Analysis of TOP1 cleavage-complexes (TOP1cc) by ICE assay. Serum-starved TDP1^{+/+} (left) and TDP1^{-/-} RPE-1 cells were treated with 25 μ M CPT for 1 h. Slot blots were incubated with MABE1084 (left) and C21 (right) anti-TOP1 antibodies. Quantification (mean \pm s.e.m.) of two independent experiments. Representative plots of TOP1cc are shown. The amount of DNA loaded is indicated..

There has been significant interest in developing TDP1 inhibitors to enhance the cytotoxicity of camptothecins and other Top1 inhibitors. The authors' results suggest that this strategy might be perilous, since it leads to Top1 mediated translocations. This possibility should be discussed.

Please see reply to concern (2) below.

Overall, the most important concerns raised here are:

(1) How does Tdp1 lead to repair of Top1 induced double strand breaks.

We think that altogether our results suggest that the removal of remaining TOP1 peptide by TDP1 is a key step in TOP1-induced DSB repair. We have openly discussed it in the new version of the manuscript. Based on your question and on a request from Reviewer2 we included in this new version ICE, repair kinetics and chromosomal translocations in PARP1^{-/-} cells. These experiments demonstrate that the accumulation of SSB and DSB is not enough to justify the defect in DSB repair and the formation of chromosomal translocations. We also detected that there is a difference between PARPi and PARP^{-/-} cells that we explain by the dominant effect of PARPi and the transient blockage of TOP1cc removal by PARPi. We can see that these results bring some new questions and bring the notion that regulation of TOP1cc processing and not only TOP1cc removal would be relevant in TOP1-induced DSB repair and in the genesis of chromosomal translocations. We hope Reviewer3 finds these new results interesting and agrees with our conclusions.

(2) If the authors results are correct, what does this suggest about the potential of Tdp1 inhibitors?

Thank you very much for this comment that highlights what is likely the most significant implication of our findings, if not the most important one. In the previous version, we indirectly addressed this subject in discussion. However, in the updated version, we have specifically and openly discussed it. We hope Reviewer3 finds this point properly addressed (453-459).

Minor points and corrections

Lines 169-170, novobiocin is a specific and potent inhibitor of POLQ; novobiocin may be potent but it is certainly not specific. Novobiocin targets bacterial topoisomerases and has weak inhibitory activity against eukaryotic type II topoisomerases. Novobiocin results led to significant confusion regarding the roles of topoisomerases in repair. I have no problem using this agent in a supporting experiment, but it is incorrect to call novobiocin specific.

Thank you very much for this comment. We have corrected the sentence (162-165). Additionally, since Reviewer2 also showed some doubts on POLQ inhibition using Novobiocin we have included experiments using a different POLQ inhibitor, ART558, that has been described as a much more specific one (Zatreanu et al Nat Comms 2021 PMID: 34140467). Please find these results in New Fig. 1F and S6.

REVIEWER COMMENTS

Reviewer #1 (Remarks to the Author):

The manuscript has been substantially improved; however, because of the lack of data using a marker-independent DSB detection, I am still not fully convinced by the present data, and researchers in the DSB repair field will likely not be convinced as well.

I have a big concern regarding the use of neutral comet assay to measure DSBs after CPT because its sensitivity must be very poor. According to the electrophoresis condition of the comet assay, only DNA fragments of 50-100 kbp or less can be detected as a tail. To overcome this limitation, PFGE assay had been developed to measure IR-induced DSBs because it is able to detect >1 Mbp fragment. In Fig. 1B, if the number of CPT-induced DSBs per G1 cell is ~10, one or fewer DSBs per chromosome are induced by CPT. If so, the size of fragmented DNA must be >1 Mbp. Therefore, I do not understand how such a large fragmented DNA is detectable by the neutral comet assay. A major concern is the contamination of S phase in arrested cells. For example, if 1% of S phase cells is contaminated in the assay, a thousand DSBs in an S phase cell significantly affects the average number of DSBs in the analyzed sample. As an important control, the authors should perform the experiment below.

In Fig. 1B, the number of 53BP1 foci per G1 cell is ~10, which is equivalent to 0.125-0.25 Gy of X-rays (or g-rays). The authors should show the result of the neutral comet assay after 0.125, 0.25, and maybe 0.5 Gy at 30 min after IR to validate the sensitivity of the comet assay.

I understand the difficulty of a sequence-based analysis. The authors do not need to perform it at this stage.

Reviewer #2 (Remarks to the Author):

The authors provided additional data and clarified some concerns. However, the major issue is whether or not TDP1 participates in a putative DSB repair pathway. Unfortunately, the authors failed to provide any evidence and/or mechanistic insight into this putative TDP1-dependent end-joining DSB repair pathway.

The data provided in this manuscript agree with the known role of TDP1 in the removal of TOP1cc especially in G0/G1 cells. TDP1 loss would lead to increased SSBs, which subsequently result in DSBs, genome instability, and cell death. Figure 2G using neutral comet assay showed that TDP1 loss led to significantly increased DSBs and some of these DSBs were repaired in TDP1^{-/-} cells (although it was not statistically significant "NS").

Additionally, there is no mechanism underlying this putative TDP1-dependent DSB repair. As shown in their working hypothesis presented in Figure 5F, all these proteins (i.e. TDP1, PARP1, LIG1/3) are well known proteins involved in SSB repair.

Reviewer #3 (Remarks to the Author):

This is a re-review of a revised manuscript by Conteres and Gomez-Herreros.

In the original ms and in this revision, they have explored how Top1 induced double strand breaks are repaired in quiescent cells. The standard model of camptothecin action, as originally proposed by Liu and co-workers, is that replication fork collisions with Top1 covalent complexes lead to the production of single-ended double strand breaks. More recent results have demonstrated that there are replication independent Top1 induced double strand breaks. The present work bears on how replication independent double strand breaks are repaired. The key finding is that in the absence of Tdp1, the repair of Top1 induced breaks can frequently lead to translocations mediated by canonical non-homologous end-joining. I think this result will be of interest to the topoisomerase community and to cancer chemotherapy investigators. Most of my comments relating to the interpretation of their results have been addressed in this revision. I still have a small number of concerns that would suggest that another round of minor revision is in order.

I am a bit troubled by two issues that were raised by the other reviewers. The first was raised by reviewer 2 regarding the generation of double strand breaks from Top1 induced ssbs. The authors need to discuss this point in greater depth. By my understanding the key piece of evidence that Tdp1 is required in a second step is the kinetics of repair with a PARP inhibitor versus when Tdp1 is absent. Might there be other factors that confound this argument? I don't think I require additional experiments, but I would like the authors to make their argument explicit in the discussion. I note some discussion of this point in lines 413-424 but I would like greater depth.

I don't think figure 5F completely captures the considerations noted in my previous paragraph. Further, I don't think that figure 5F or the text makes clear what Tdp1 is doing in double strand break repair. This is in accord with the point raised by reviewer 1. I think the present work is of sufficient significance, but I would like the authors to address this question in greater depth

Minor points

I think the authors need to be careful relating their work on repair of etoposide induced damage to repair of Top1 damage. This was mainly raised in responses to reviewers, but appropriate caveats are needed.

I was surprised that Tdp2 was not mentioned at all.

Lines 48-50. Current literature notes that other proteases likely participate in proteolyzing trapped Top1.

Line 57 Phosphatase is mis-spelled.

Line 164 should read or ART558 not and ART558 (I assume the authors did not treat with both at the same time).

Lines 381-386 Regarding potential roles of RNA pol I in inducing double strand breaks, I wonder if the assays used have the requisite sensitivity. I would prefer that the conclusion that Top1 induced dsbs are independent of Pol1 be downplayed unless appropriate controls demonstrate that they would be detectable.

Lines 393-394 Please rewrite for clarity.

Lines 431-435 Please rewrite for clarity. Conditionate is an obsolete word.

REVIEWER COMMENTS

Reviewer #1 (Remarks to the Author):

The manuscript has been substantially improved; however, because of the lack of data using a marker-independent DSB detection, I am still not fully convinced by the present data, and researchers in the DSB repair field will likely not be convinced as well. I have a big concern regarding the use of neutral comet assay to measure DSBs after CPT because its sensitivity must be very poor. According to the electrophoresis condition of the comet assay, only DNA fragments of 50-100 kbp or less can be detected as a tail. To overcome this limitation, PFGE assay had been developed to measure IR-induced DSBs because it is able to detect >1 Mbp fragment. In Fig. 1B, if the number of CPT-induced DSBs per G1 cell is ~10, one or fewer DSBs per chromosome are induced by CPT. If so, the size of fragmented DNA must be >1 Mbp. Therefore, I do not understand how such a large fragmented DNA is detectable by the neutral comet assay. A major concern is the contamination of S phase in arrested cells. For example, if 1% of S phase cells is contaminated in the assay, a thousand DSBs in an S phase cell significantly affects the average number of DSBs in the analyzed sample. As an important control, the authors should perform the experiment below.

In Fig. 1B, the number of 53BP1 foci per G1 cell is ~10, which is equivalent to 0.125-0.25 Gy of X-rays (or g-rays). The authors should show the result of the neutral comet assay after 0.125, 0.25, and maybe 0.5 Gy at 30 min after IR to validate the sensitivity of the comet assay. I understand the difficulty of a sequence-based analysis. The authors do not need to perform it at this stage.

We totally agree with Reviewer1 about the low sensitivity of neutral comets. We indicated this limitation in our previous response. We find his/her suggestion very useful to calibrate the sensitivity of the technique. Nevertheless, we would also like to indicate that:

1. The number of breaks after 1 hour treatment with 25µM CPT is not 10 but between 10 and 20 in wild-type cells and around 40-50 in TDP1^{-/-} cells. Therefore, there is still a small window to detect DSBs induced by CPT.

2. An effect of remaining cycling cells is possible, but it would correspond to less than 2% (Figure S1) of cells. We think that the fact that neutral comet assay is a single-cell approach prevents this 2% affecting the result. To prove it we have compared the distribution of comet tails among cell population +/- CPT in WT and TDP1^{-/-} cells (please see Figure R1 below). The percentage of cells that increase their tail moment is much higher than 2% indicating that the increase is not restricted to S-phase cells but to the general population.

Figure R1. Detection of DSBs by neutral comet assay in serum-starved TDP1^{+/+} (left) and TDP1^{-/-} (right) RPE-1 cells untreated or following treatment with CPT (25 µM) for 1 h and after 2 h repair in drug-free medium. Distribution (relative frequency in percentage) of comet tail moment is shown. We think that the fact that we detect the formation of chromosomal translocations induced by PCC upon CPT treatment is a direct, marker-independent and unambiguous demonstration of the

formation of TOP1-induced DSBs. Nevertheless, we considered the suggestion of Reviewer1 of doing PFGE, but in case there were an effect of remaining <2% of cycling cells we thought it would also affect a PFGE-based approach so it would not solve this technical limitation. Finally, to respond to Reviewer1 request we have now measure directly DSBs formation upon CPT treatment by PCC-Giemsa (New Figure S6B, text 296-299). With this approach we have directly visualized chromosomal breakage induced by CPT in G0 cells. These results demonstrate that CPT induces DSBs in quiescent cells and that TDP1-/- cells accumulate them. We hope that Reviewer 1 and editor find this new result conclusive.

Reviewer #3 (Remarks to the Author):

This is a re-review of a revised manuscript by Conteres and Gomez-Herreros. In the original ms and in this revision, they have explored how Top1 induced double strand breaks are repaired in quiescent cells. The standard model of camptothecin action, as originally proposed by Liu and co-workers, is that replication fork collisions with Top1 covalent complexes lead to the production of single-ended double strand breaks. More recent results have demonstrated that there are replication independent Top1 induced double strand breaks. The present work bears on how replication independent double strand breaks are repaired. The key finding is that in the absence of Tdp1, the repair of Top1 induced breaks can frequently lead to translocations mediated by canonical non-homologous end-joining. I think this result will be of interest to the topoisomerase community and to cancer chemotherapy investigators. Most of my comments relating to the interpretation of their results have been addressed in this revision. I still have a small number of concerns that would suggest that another round of minor revision is in order.

We thank Reviewer3 for his/her support and constructive suggestions that have helped us improving the clarity and accuracy of our discussion.

I am a bit troubled by two issues that were raised by the other reviewers. The first was raised by reviewer 2 regarding the generation of double strand breaks from Top1 induced ssbs. The authors need to discuss this point in greater depth. By my understanding the key piece of evidence that Tdp1 is required in a second step is the kinetics of repair with a PARP inhibitor versus when Tdp1 is absent. Might there be other factors that confound this argument? I don't think I require additional experiments, but I would like the authors to make their argument explicit in the discussion. I note some discussion of this point in lines 413-424 but I would like greater depth.

We are not completely sure if we have totally achieved the depth of discussion Reviewer3 demands. We hope we have this time. Please find discussion regarding PARPi, PARP1-/- and TDP1-/- in 411-434.

I don't think figure 5F completely captures the considerations noted in my previous paragraph. Further, I don't think that figure 5F or the text makes clear what Tdp1 is doing in double strand break repair. This is in accord with the point raised by reviewer 1. I think the present work is of sufficient significance, but I would like the authors to address this question in greater depth.

We have tried to generate a basic model including only those factors incorporated in this study. The main objective of our model was to underline the key role of TOP1 adduct removal by TDP1 in the repair of TOP1-induced DSBs. And, more importantly, that TDP1 prevents chromosomal translocations. We are aware that many other factors and steps, and a much higher degree of complexity could be included. Regarding PARP1 activity, we could have included unprocessed TOP1 upstream of TDP1 activity, and PARP1 role upstream of TDP1. However, we considered that keeping it simple would benefit the reader. As discussed above, we have extended discussion on this point, and we hope this will clarify the view of the model as well. We are open to modify the model in case that Reviewer3 consider it adequate.

Minor points

I think the authors need to be careful relating their work on repair of etoposide induced damage to repair of Top1 damage. This was mainly raised in responses to reviewers, but appropriate caveats are needed.

We are very sorry, but we are not completely sure about what Reviewer3 suggests for us to amend here. We take etoposide-induced DSB repair as a reference of LIG4-/- repair defect, but we suggest about the singularity of TOP1-induced DSBs underlying the difference between them. 382-388.

I was surprised that Tdp2 was not mentioned at all.

We have added TDP2 and two new citations to discussion. 515-522

Lines 48-50. Current literature notes that other proteases likely participate in proteolyzing trapped Top1.

We have added this to introduction. 49-50

Line 57 Phosphatase is mis-spelled.

This has now been corrected. 58.

Line 164 should read or ART558 not and ART558 (I assume the authors did not treat with both at the same time).

This has now been corrected. 165.

Lines 381-386 Regarding potential roles of RNA pol I in inducing double strand breaks, I wonder if the assays used have the requisite sensitivity. I would prefer that the conclusion that Top1 induced dsbs are independent of Pol1 be downplayed unless appropriate controls demonstrate that they would be detectable.

This part of the discussion has been removed.

Lines 393-394 Please rewrite for clarity.

This part has now been rewritten. 391-394

Lines 431-435 Please rewrite for clarity. Conditionate is an obsolete word.

Conditionate has been removed and this part has now been rewritten. 435-452

REVIEWERS' COMMENTS

Reviewer #1 (Remarks to the Author):

The authors adequately answered my concerns.

Reviewer #3 (Remarks to the Author):

This is a re-review of a revised manuscript by Conteres and Gomez-Herreros. In the original ms and in their revisions, the authors have explored how Top1 induced double strand breaks are repaired in quiescent cells. The standard model of camptothecin action, as originally proposed by Liu and co-workers, is that replication fork collisions with Top1 covalent complexes lead to the production of single-ended double strand breaks. More recent results have demonstrated that there are replication independent Top1 induced double strand breaks. The present work bears on how replication independent double strand breaks are repaired. The key finding is that in the absence of Tdp1, the repair of Top1 induced breaks can frequently lead to translocations mediated by canonical non-homologous end-joining. I think this result will be of interest to the topoisomerase community and to cancer chemotherapy investigators. All of my comments relating to the interpretation of their results have been addressed in this revision. The authors have also satisfactorily addressed all of the minor points raised in the original reviews.